# Exploration for Building Next-Generation Foundation MLLMs via Self-Learning

## ABSTRACT

While inference-time computation and post-training optimization have significantly advanced multimodal large language models (MLLMs), these advancements remain constrained by the capabilities of foundation models. We argue that effective model advancement requires strong synergy among pre-training, inference-time computation, and post-training optimization. In this paper, we introduce **S**elf-**I**mproving **cog**nition (SICOG), a self-learning framework for building next-generation foundation MLLMs by imparting multimodal knowledge and enhancing systematic cognitive capabilities through multimodal pre-training with self-generated data. Specifically, we propose *Chain-of-Description* for step-by-step visual understanding and integrate structured Chain-of-Thought (CoT) reasoning to support in-depth multimodal reasoning. SICOG first equips a base model with systematic perception and reasoning using minimal external supervision. The enhanced models then generate candidate image captions and CoT reasoning responses for ***unlabeled*** images and image-question pairs across diverse tasks, which are filtered through a semantic-similarity-guided self-consistency mechanism. These high-quality, self-generated samples enable large-scale multimodal pre-training, creating a self-improvement loop. Experiments demonstrate SICOG's effectiveness in developing MLLMs with enhanced multimodal cognition. Using only 213K self-generated pre-training samples, SICOG achieves significant improvements, including +3.6% on MMStar and +3.5% on AI2D, outperforming previous pre-training approaches. When combined with post-training techniques for CoT reasoning, SICOG yields +9% gains on MMVet and +8.5% on ScienceQA.

## 1 INTRODUCTION

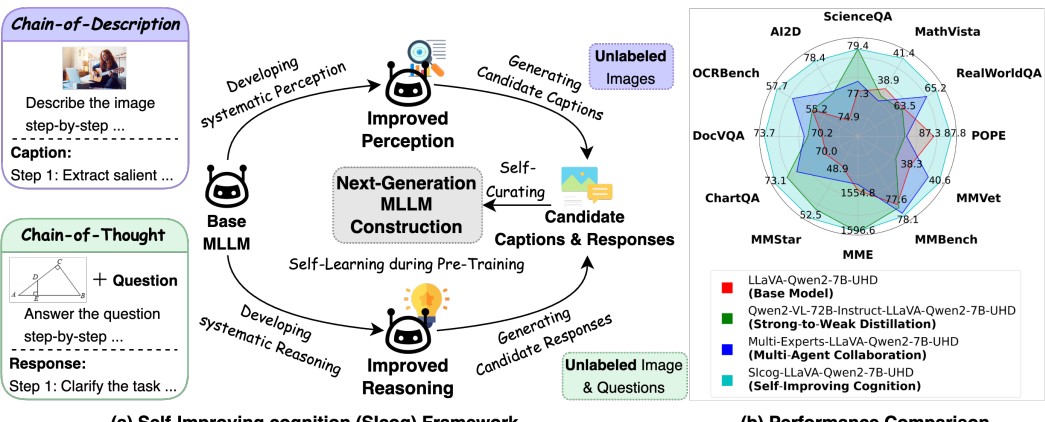

(a) **S**elf-**I**mproving **cog**nition (SIcog) Framework      (b) **Performance Comparison**

Figure 1: (a) SICOG enhances an MLLM's systematic cognition during multimodal pre-training using self-generated data, enabling next-generation foundation MLLMs. (b) With up to 213K self-generated pre-training samples, SICOG produces foundation MLLMs with superior cognitive capabilities, showing benchmark-leading performance compared to prevalent pre-training approaches.

Recent efforts in inference-time computation (Teng et al., 2025) and post-training optimization (Guo et al., 2025; Feng et al., 2025) have significantly enhanced the capabilities of MLLMs (Yang et al., 2024b; Bai et al., 2023), particularly in areas such as multimodal reasoning (Xu et al., 2025). However, these advancements remain largely constrained by the foundational knowledge and capabilities of the models (Shah, 2024; Liu et al., 2025; Gandhi et al., 2025), which are determined during pre-training (Sutskever, 2024). **We argue that effective model advancement requires a strong synergy between pre-training and downstream mechanisms, such as inference-time computation and post-training optimization.** Pre-training provides the essential foundation, and its seamless integration with downstream processes is crucial for achieving robust and scalable performance.

In this paper, we focus on the effective advancement of foundation MLLMs (Li et al., 2024a), a critical step toward enabling real-world understanding (Bordes et al., 2024). Prevalent multimodal pre-training approaches for foundation MLLM construction (Chen et al., 2024a; Fang et al., 2024) typically rely on large-scale training with high-quality image–caption data generated by advanced MLLMs (OpenAI, 2023; Liu et al., 2024b) to equip models with diverse multimodal knowledge and fine-grained visual perception skills (Deng et al., 2024; Sun et al., 2024). Nonetheless, these generated captions often lack comprehensiveness and accuracy. To improve coverage, some methods incorporate fine-grained attributes using annotations from multiple expert models (Sun et al., 2024; Fang et al., 2024). However, the resulting captions often lack fluency and coherence due to the absence of an underlying logical structure. Moreover, these pre-training approaches tend to neglect the development of multimodal reasoning capabilities (Xu et al., 2025), which are essential for extending the practical utility of MLLMs in real-world applications (Li et al., 2024b).

Inspired by human experiential learning (*a.k.a.* human cognitive development) (Khatun et al., 2023; Silver & Sutton, 2025), we introduce **SICOG**, a self-learning framework that imparts multimodal knowledge and enhances MLLMs' systematic cognitive abilities—including both perception and reasoning—during multimodal pre-training with self-generated data for next-generation foundation MLLM construction. Central to SICOG is *Chain-of-Description* (CoD), which enables an MLLM to interpret visual content through structured, step-by-step analysis. CoD sequentially focuses on four critical aspects—*salient content, fine-grained details, relational attributes*, and *peripheral context*—before generating a coherent and logically grounded description. This design ensures comprehensive coverage while mitigating hallucinations. We further incorporate structured CoT reasoning (Xu et al., 2025), which has been shown to significantly enhance complex reasoning by enabling in-depth multimodal analysis prior to answer generation and fostering coherent integration of visual and textual information. As illustrated in Figure 1 (left), SICOG first develops an MLLM's systematic perceptual and reasoning capabilities using minimal external supervision. This is achieved by fine-tuning a base model on a small set of high-quality caption data enriched with our proposed *Chain-of-Description*, along with a limited amount of structured CoT reasoning data (**post-training optimization**). The fine-tuned model then generates multiple candidate captions and responses for ***unlabeled multimodal data across diverse tasks***. To avoid dependence on external models, we apply a self-consistency mechanism (Wang et al., 2022; Wu et al., 2024) to curate these self-generated outputs, selecting higher-quality samples based on semantic coherence (**inference-time computation**). Finally, the curated data are used for large-scale multimodal **pre-training**, completing a self-learning cycle, resulting in a more capable, cognitively grounded foundation MLLM.

Following Korbak et al. (2023), we prioritize the comparison with various pre-training approaches. Specifically, we evaluate SICOG on both low-resolution and high-resolution MLLMs across diverse benchmarks. Extensive experimental results (Figure 1, right) demonstrate that SICOG produces stronger foundation models with enhanced multimodal cognition, significantly outperforming prevalent pre-training methods (Li et al., 2024a; Fang et al., 2024). In addition, SICOG enhances post-training performance and promotes continual self-improvement in newly constructed models. In summary, our contributions are three-fold:

- We propose SICOG, a self-learning framework that enhances MLLMs' systematic cognition for constructing next-generation foundation MLLMs through multimodal pre-training with self-generated data.
- We introduce *Chain-of-Description*, a structured visual understanding mechanism that enables step-by-step interpretation of visual content to improve perceptual quality.
- We demonstrate SICOG's effectiveness across various benchmarks on both low- and high-resolution MLLMs, significantly surpassing previous approaches.

## 2 RELATED WORK

**Multimodal Pre-Training.** Multimodal (vision-language) pre-training has proven highly effective in imparting multimodal knowledge and enhancing the perceptual capabilities of MLLMs by leveraging diverse, high-quality image–caption datasets (Lu et al., 2024a; Bai et al., 2023; Liu et al., 2024b). However, the construction of such datasets often depends on proprietary or open-source models to generate detailed captions (Chen et al., 2024a; Li et al., 2024d), or on expert visual models to extract fine-grained attributes (Peng et al., 2023), which are subsequently integrated into descriptive captions (Fang et al., 2024; Xu et al., 2024a; Sun et al., 2024). To reduce reliance on external annotations, we leverage the model's inherent visual instruction-following and generalization capabilities to generate detailed caption data for self-improvement, similar to Fang et al. (2024); Deng et al. (2024). Beyond perception, we further enhance the model's multimodal reasoning abilities by incorporating self-generated visual instruction-tuning data, including both direct answers and CoT formats. This enables a shift from focusing solely on perception to advancing cognitive capabilities. Detailed discussion is provided in Appendix D.

## 3 METHODOLOGY: THE SICOG FRAMEWORK

In this section, we introduce SICOG, a self-learning framework for constructing next-generation foundation MLLMs. We begin with a comprehensive overview in Section 3.1, then delve into the details of *Chain-of-Description* for systematic perception enhancement in Section 3.2, followed by structured CoT for systematic reasoning enhancement in Section 3.3. A comprehensive introduction to SICOG is available in Appendix B.

### 3.1 OVERVIEW

The goal of SICOG is to advance MLLMs by equipping them with rich multimodal knowledge and systematic cognitive capabilities—namely, systematic visual understanding ("how to observe") and in-depth multimodal reasoning ("how to think")—during multimodal pre-training, with minimal reliance on external annotations. As illustrated in Figure 2, SICOG operates through four key steps.

**Step 1: Developing Systematic Multimodal Cognition with Minimal Annotations.** We enhance the MLLM, $\mathcal{M}$ (parameterized by $\theta$), by fine-tuning it to systematically interpret and integrate multimodal information using minimal annotated data. This includes two main components:

- **Systematic multimodal perception.** To improve the MLLM's ability to systematically observe and interpret images, we fine-tune $\mathcal{M}$ using minimal high-quality image-captioning datasets $\mathcal{D}^{\text{Perception}}$, resulting in an enhanced perception model, $\mathcal{M}_0^{\text{Perception}}$. These datasets include images $v$, prompts $x$, step-by-step analyses $s$, and descriptions $y$, structured in two formats: Detailed Description (DD) and *Chain-of-Description* (CoD). Details of the *Chain-of-Description* strategy and data collection are provided in Section 3.2.

$$\mathcal{D}^{\text{Perception}} = \mathcal{D}_{DD}^{\text{Perception}} + \mathcal{D}_{CoD}^{\text{Perception}} = \{(v_i, x_i, y_i)\}_{i=1}^N + \{(v_i, x_i, s_i, y_i)\}_{i=1}^N, \quad (1)$$

where $N$ is the number of training samples.

- **Systematic multimodal reasoning.** To improve reasoning, we fine-tune $\mathcal{M}$ with minimal visual instruction-tuning datasets $\mathcal{D}^{\text{Reasoning}}$, resulting in $\mathcal{M}_0^{\text{Reasoning}}$. These datasets include images $v$, questions $q$, intermediate step-by-step rationales $r$, and answers $a$, structured as Direct Answer (DA) and Chain-of-Thought (CoT). Details of data curation are provided in Section 3.3.

$$\mathcal{D}^{\text{Reasoning}} = \mathcal{D}_{DA}^{\text{Reasoning}} + \mathcal{D}_{CoT}^{\text{Reasoning}} = \{(v_i, q_i, a_i)\}_{i=1}^M + \{(v_i, q_i, r_i, a_i)\}_{i=1}^M, \quad (2)$$

where $M$ is the number of samples.

**Step 2: Generating Candidate Captions and Responses for Pre-Training Data Collection.** We construct multimodal pre-training data by leveraging the improved models, $\mathcal{M}_0^{\text{Perception}}$ and $\mathcal{M}_0^{\text{Reasoning}}$, to generate candidate image captions and visual instruction responses. This step involves:

- **Image caption candidate generation.** Given a set of *unlabeled* images $\{v_k\}_{k=1}^K$, we prompt $\mathcal{M}_0^{\text{Perception}}$ (with policy $p_{\mathcal{M}_0^{\text{Perception}}}$) using two types of instructions to generate detailed descriptions and induce *Chain-of-Description* perception:
  1. "Please generate a detailed caption of this image." ($x_{DD}$).

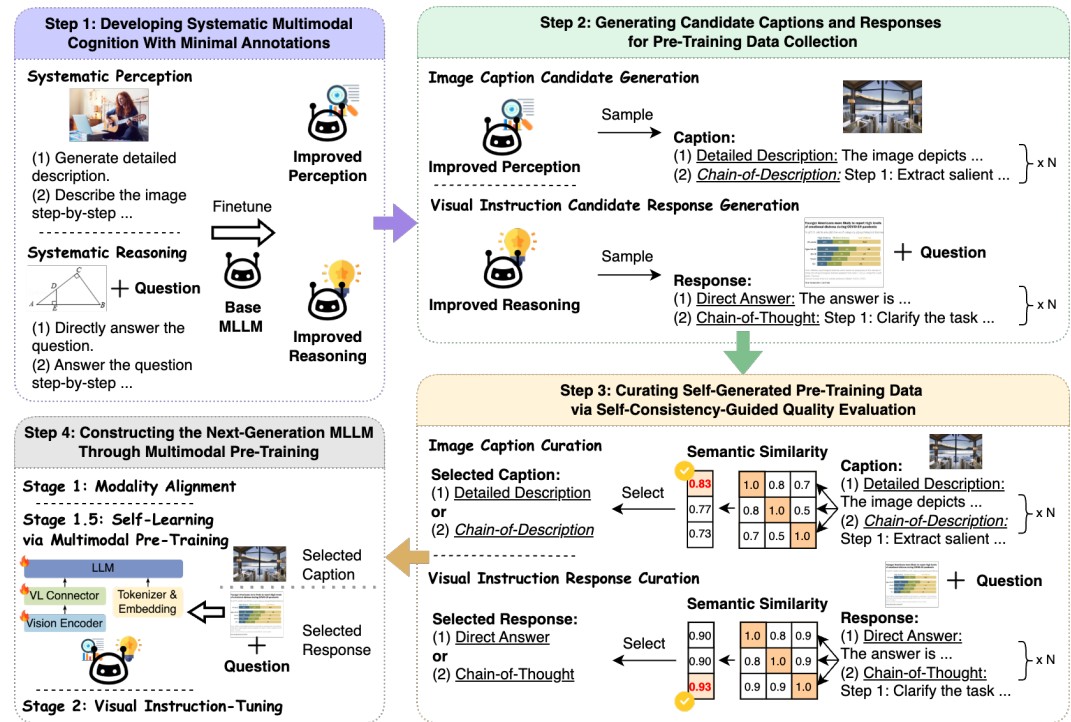

Figure 2: The SICOG framework comprises four steps: (*i*) Developing multimodal cognitive capabilities by finetuning an MLLM with minimal annotated image-captioning data (with *Chain-of-Description*) and visual instruction-tuning data (with structured Chain-of-Thought), enhancing systematic perception and reasoning (upper left). (*ii*) Generating candidate captions and responses for pre-training by sampling from the improved models (upper right). (*iii*) Curating self-generated pre-training data through self-consistency-guided quality evaluation, selecting the most semantically consistent candidates for learning (lower right). (*iv*) Constructing a next-generation foundation MLLM by performing multimodal pre-training on the curated data (lower left). For brevity, language ability preservation is omitted; see Figure 3 for the complete version.

2. "Please generate ... Describe the image step by step." ($x_{CoD}$).

For each image $v_k$, $\mathcal{M}_0^{\text{Perception}}$ generates multiple candidate captions via sampling:

$$\{\hat{y}_k\} \sim p_{\mathcal{M}_0^{\text{Perception}}}(\cdot \mid v_k, x_{DD}), \{(\hat{s}_k, \hat{y}_k)\} \sim p_{\mathcal{M}_0^{\text{Perception}}}(\cdot \mid v_k, x_{CoD}), \quad (3)$$

where $\{\hat{y}_k\}$ is the set of detailed descriptions, and $\{(\hat{s}_k, \hat{y}_k)\}$ is the set of step-by-step analyses with descriptions. The resulting dataset includes captions in two formats.

- **Visual instruction candidate response generation.** For a set of *unlabeled* images $\{v_z\}_{z=1}^{Z}$ with corresponding questions $q_z$, we prompt $\mathcal{M}_0^{\text{Reasoning}}$ (with policy $p_{\mathcal{M}_0^{\text{Reasoning}}}$) using two types of instructions to generate direct answers and induce Chain-of-Thought reasoning:

  1. "<original question>." ($q_{DA}$).
  2. "<original question> Answer the question step by step." ($q_{CoT}$).

For each image $v_z$ and question $q_z$, $\mathcal{M}_0^{\text{Reasoning}}$ produces multiple candidate responses:

$$\{\hat{a}_z\} \sim p_{\mathcal{M}_0^{\text{Reasoning}}}(\cdot \mid v_z, q_{DA}), \{(\hat{r}_z, \hat{a}_z)\} \sim p_{\mathcal{M}_0^{\text{Reasoning}}}(\cdot \mid v_z, q_{CoT}), \quad (4)$$

where $\{\hat{a}_z\}$ is the set of direct answers, and $\{(\hat{r}_z, \hat{a}_z)\}$ is the set of step-by-step rationales with answers. The resulting dataset includes responses in two formats.

**Step 3: Curating Self-Generated Pre-Training Data via Self-Consistency-Guided Quality Evaluation.** To ensure the quality of self-generated pre-training data across diverse tasks, we employ *self-consistency* (Wu et al., 2024; Li et al., 2024c) to evaluate candidate samples without external supervision. This method is based on the principle that *higher-quality candidates exhibit*

*greater semantic consistency*. The most consistent candidates are selected for further self-refinement during multimodal pre-training.

Specifically, we apply a semantic-similarity-guided self-consistency evaluation function, $f(\cdot)$. For each instance (*e.g.,* an *unlabeled* image), it assesses the quality of candidates (*e.g.,* candidate captions) by comparing each candidate against all others based on semantic similarity and selects the candidate with the highest consistency score, provided it exceeds a predefined threshold $\tau$ (*i.e.,* otherwise, the instance and its candidates are skipped):

$$f(\{c\}) = \arg\max_{c \in \{c\}} \frac{1}{N_{\text{cand}}} \sum_{j=1}^{N_{\text{cand}}} \text{sim}(c, c^{(j)}), \quad s.t. \quad \max_{c \in \{c\}} \frac{1}{N_{\text{cand}}} \sum_{j=1}^{N_{\text{cand}}} \text{sim}(c, c^{(j)}) \geq \tau, \quad (5)$$

where $N_{\text{cand}}$ is the number of candidate samples being compared, and $\{c\}$ represents the candidate set. As illustrated in the lower-right part of Figure 2, we apply this method as follows:

- **Image caption curation.** For each image $v_k$, we apply $f(\cdot)$ to evaluate the quality of candidate captions by comparing each generated description $\hat{y}_k$ against all others. The caption with the highest semantic consistency is selected as the final self-generated caption, resulting in a curated dataset of selected captions $\mathcal{D}_{\text{Selected}}^{\text{Perception}}$.

- **Visual instruction response curation.** For each image $v_z$ and question $q_z$, $f(\cdot)$ evaluates candidate responses by comparing each generated answer $\hat{a}_z$ against all others. The most consistent response is selected, resulting in a curated dataset $\mathcal{D}_{\text{Selected}}^{\text{Reasoning}}$.

In addition, to preserve language capabilities, we prompt the backbone LLM, $\mathcal{M}_{LLM}$, to generate candidate responses for *unlabeled* text-only instructions. These responses are then curated using a similar process, resulting in $\mathcal{D}_{\text{Selected}}^{\text{Language}}$. Combining all three curated datasets yields the final self-generated pre-training dataset $\mathcal{D}^{\text{Pre-training}}$.

**Step 4: Constructing the Next-Generation MLLM through Multimodal Pre-Training.** To build the next-generation foundation MLLM, we introduce an intermediate multimodal pre-training stage, Stage 1.5, within the standard two-stage training strategy, following (Liu et al., 2024b; Li et al., 2024a). This stage improves the MLLM using curated self-generated pre-training data. The complete training strategy consists of three stages, as shown in the lower left part in Figure 2: (*i*) **Stage 1: Modality alignment.** Align image features with the text embedding space. Following (Li et al., 2024a), only the vision-language connector is trained on image-text pairs during this stage. (*ii*) **Stage 1.5: Self-learning via multimodal pre-training.** Train the model on curated pre-training data $\mathcal{D}^{\text{Pre-training}}$ to acquire multimodal knowledge and integrate systematic perception and reasoning. During this stage, all model components are fully trainable. (*iii*) **Stage 2: Visual instruction-tuning.** Fine-tune the model on instruction-tuning data to develop robust visual instruction-following capabilities. All model components remain fully trainable.

## 3.2 ENHANCING SYSTEMATIC PERCEPTION WITH *Chain-of-Description*

We introduce *Chain-of-Description* (CoD) to enable systematic perception, equipping the MLLM with the ability to logically analyze and describe visual information step by step ("how to observe"). This step-by-step approach enhances thorough visual interpretation. As shown in Figure 4 (left), *Chain-of-Description* consists of five sequential stages: (*i*) **Step 1: Extract salient content.** Identify the key elements that define the overall context and meaning of the image, laying the foundation for basic visual recognition. (*ii*) **Step 2: Analyze detailed information.** Focus on instance-level attributes, such as low-level and fine-grained details, *e.g.,* "the guitar is a classic wooden brown with light-colored frets." This step ensures a precise and detailed interpretation of the image. (*iii*) **Step 3: Consider relational-level attributes.** Analyze interactions between elements and their spatial organization, *e.g.,* "the person is seated on the bed," leading to a richer and more comprehensive understanding of visual relationships. (*iv*) **Step 4: Examine marginal or peripheral content.** Pay attention to less prominent or background details, *e.g.,* "the dresser in the background," to ensure no important information is overlooked. (*v*) **Step 5: Organize all observations.** Synthesize all findings into a cohesive, detailed description, enabling comprehensive coverage and holistic image understanding. Due to space constraints, details regarding the **data preparation** for minimally annotated CoD data are provided in Appendix B.1.

## 3.3 Improving Systematic Reasoning with Structured Chain-of-Thought

We adopt structured CoT (Xu et al., 2025) to enhance MLLMs' systematic reasoning capabilities. As shown in Figure 4 (right), this approach decomposes complex multimodal tasks into logical steps, enabling progressive analysis and reasoning. The structured CoT process follows four key stages: (*i*) **Step 1: Clarify the task objective.** Identify the problem's requirements to establish a foundational understanding. (*ii*) **Step 2: Extract crucial visual information.** Identify relevant visual elements to inform the reasoning process. (*iii*) **Step 3: Generate detailed reasoning.** Construct a logical chain of intermediate steps based on the visual and textual context. (*iv*) **Step 4: Conclude with an answer.** Synthesize the reasoning steps into a coherent and accurate response. Due to space limitations, details on the **data preparation** for minimally annotated CoT data are provided in Appendix B.2.

## 4 Experiments

### 4.1 Experimental Setup

**Datasets and Evaluation Metrics.** We evaluate the efficacy of SICOG on the following well-established benchmarks, using the open-source evaluation toolkit VLMEvalKit (Duan et al., 2024): (*i*) **Multimodal Comprehensive Understanding:** MMStar (Chen et al., 2024c), MMBench (Liu et al., 2024d), MMVet (Yu et al., 2024b) (report accuracy). (*ii*) **Hallucination:** POPE (Li et al., 2023) (report F1 score). (*iii*) **Chart/Table Understanding:** OCRBench (Liu et al., 2024e), DocVQA (Tito et al., 2021), ChartQA (Masry et al., 2022) (report accuracy). (*iv*) **Knowledge-Oriented Tasks:** MathVista (Lu et al., 2024b), ScienceQA (Lu et al., 2022), AI2D (Kembhavi et al., 2016) (report accuracy). (*v*) **Real-World Understanding:** RealWorldQA (X.AI, 2024) (report accuracy).

**Compared Methods.** We compare SICOG against the following representative MLLM pre-training approaches (as discussed in Section 2). Differences are considered significant at $p < 0.01$: (*i*) **Strong-to-Weak Distillation (Perception)** (Li et al., 2024a): Pre-training with re-caption data containing detailed descriptions (DD) generated by stronger models. (*ii*) **Multi-Agent Collaboration (Perception)** (Fang et al., 2024): Pre-training with re-caption data containing detailed descriptions and fine-grained attributes (DD-FGA) generated by base and expert models. Due to space limitations, we provide **Implementation Details** in Appendix M.

### 4.2 Can Self-Improved Systematic Cognition Yield Next-Generation Foundation MLLMs?

Table 1 presents the comprehensive evaluation results. Following Korbak et al. (2023), we prioritize comparisons with various pre-training approaches rather than emphasizing state-of-the-art (SOTA) performance. The key observations are summarized as follows:

**SICOG yields next-generation foundation MLLMs with self-improved cognitive capabilities.** SICOG consistently improves both high-resolution and low-resolution foundation MLLMs across diverse tasks, achieving gains of 2%–3.5% on MMStar for comprehensive tasks, 2%–3% on perception tasks (*e.g.,* DocVQA and ChartQA), and 2%–4% on reasoning tasks (*e.g.,* ScienceQA and AI2D).

**Systematic perception through self-learning strengthens foundation MLLMs.** SICOG (Perception), which leverages self-generated captions with detailed descriptions and *Chain-of-Description*, achieves comparable or superior accuracy across benchmarks relative to strong-to-weak distillation and multimodal collaboration methods. Unlike these alternatives, which heavily rely on extensive external annotations, SICOG reduces this dependence through self-learning.

**Integrating systematic reasoning into pre-training proves more effective than prioritizing perception alone.** SICOG (Perception + Reasoning) boosts multimodal reasoning, surpassing perception-only methods by 2.5%–4% on ScienceQA while preserving strong perception capabilities. Notably, perception-only pre-training degrades performance on hallucination tasks (a 0.5%–1% drop on POPE), whereas systematic reasoning mitigates this issue and maintains robustness. Incorporating self-generated text-only instruction-tuning data during pre-training further enhances performance, especially for high-resolution MLLMs. This observation aligns with findings in (Lu et al., 2024a).

**Stronger foundation MLLMs enable more effective self-improvement.** SICOG achieves greater performance gains when applied to LLaVA-Qwen2-7B-UHD (higher baseline capabilities) compared to LLaVA-Llama3.1-8B-UHD (lower baseline capabilities), showing that base model performance significantly influences self-improvement potential, with stronger MLLMs yielding better results.

Table 1: Evaluation results on eight benchmarks (direct answer inference). *The only difference between the compared methods is the pre-training data utilized in Stage 1.5 (see details in Step 4 of Section 3).* Results marked with an asterisk (∗) are cited from the OpenVLM Leaderboard (Duan et al., 2024). Some results are provided in Appendix P.

| Method | Train Data | | Comprehensive | | Hallu. | Chart/Table | | Knowledge | | |
|---|---|---|---|---|---|---|---|---|---|---|
| | Stage 1.5 | | MMBen. | MMStar | POPE | DocV. | Chart. | Math. | Science. | AI2D |
| *Open-Sourced Models (For holistic analysis, not for comparison)* | | | | | | | | | | |
| VITA-1.0-Mixtral-8x7B∗ (Fu et al., 2024a) | - | | - | 46.60 | - | - | - | 44.50 | - | 72.80 |
| LLaVA-v1.5-13B∗ (Liu et al., 2024a) | - | | 69.20 | 34.30 | 88.40 | - | 18.20 | 27.70 | 72.60 | 61.10 |
| ShareGPT4V-13B∗ (Chen et al., 2024b) | - | | 69.80 | 38.30 | 87.50 | - | 24.60 | 29.40 | 72.60 | 61.40 |
| Molmo-7B-O∗ (Deitke et al., 2024) | - | | 72.20 | 50.10 | 86.70 | - | 36.50 | 43.90 | 88.80 | 75.70 |
| Eagle-X5-13B∗ (Shi et al., 2024) | - | | 72.60 | 43.70 | 89.80 | - | 69.60 | 40.80 | 71.80 | 77.00 |
| CogVLM2-19B-Chat∗ (Chen et al., 2025) | - | | 73.90 | 50.50 | 83.40 | - | 33.00 | 38.70 | 90.20 | 73.40 |
| LLaVA-NeXT-8B∗ (Li et al., 2024a) | - | | 74.80 | 43.90 | 87.10 | - | 68.70 | 37.70 | 73.10 | 72.80 |
| Cambrian-1-8B∗ (Tong et al., 2024) | - | | 74.60 | 50.70 | 86.40 | - | 72.60 | 48.10 | 81.00 | 74.60 |
| XGen-MM-Instruct-Interleave-v1.5∗ (Xue et al., 2024) | - | | 78.30 | 48.40 | 87.20 | - | - | 40.60 | 88.30 | 74.20 |
| Janus-Pro-7B∗ (Chen et al., 2025) | Sufficient high-quality multimodal data | | 62.60 | 46.50 | 78.90 | - | - | 42.50 | 83.20 | 68.10 |
| DeepSeek-VL-7B∗ (Lu et al., 2024a) | Sufficient high-quality multimodal data | | 73.80 | 40.50 | 85.60 | - | 59.10 | 37.20 | 80.90 | 65.30 |
| VILA1.5-13B∗ (Lin et al., 2024) | Sufficient high-quality multimodal data | | 74.40 | 44.20 | 85.00 | - | - | 42.30 | 79.10 | 69.90 |
| **Low-Resolution** | | | | | | | | | | |
| *Base Model* | | | | | | | | | | |
| LLaVA-Qwen2-7B (Liu et al., 2023) | - | | 74.44 | 46.67 | **84.55** | 50.62 | 52.72 | 38.00 | 74.91 | 73.77 |
| *Strong-to-Weak Distillation (Perception)* | | | | | | | | | | |
| LLaVA-NeXT-34B-LLaVA-Qwen2-7B | 118k Caption w/ DD by LLaVA-NeXT-34B | | **76.18** | 46.73 | 83.72 | 51.26 | 52.68 | 36.60 | 75.56 | 74.38 |
| Qwen2-VL-72B-Instruct-LLaVA-Qwen2-7B | 118k Caption w/ DD by Qwen2-VL-72B-Instruct | | 75.84 | 48.20 | 83.84 | 50.85 | 52.56 | 36.10 | 76.15 | 74.87 |
| *Multi-Agent Collaboration (Perception)* | | | | | | | | | | |
| Multi-Experts-LLaVA-Qwen2-7B | 118k Caption w/ DD-FGA by base and expert models | | 76.01 | 47.60 | 84.12 | 51.06 | 53.36 | **38.90** | 75.46 | 74.97 |
| *Self-Improving Cognition (Perception & Reasoning)* | | | | | | | | | | |
| SICOG-LLaVA-Qwen2-7B (Perception) | Self-generated 118k caption w/ DD&CoD | | 75.34 | 48.27 | 83.89 | 50.83 | 54.88 | 38.50 | 74.71 | 75.13 |
| SICOG-LLaVA-Qwen2-7B (Perception, Reasoning) | Self-generated 118k caption w/ DD&CoD, 45k VQA w/ DA&CoT | | 76.01 | **48.67** | 84.10 | **52.70** | **55.20** | 38.10 | **78.88** | 76.78 |
| SICOG-LLaVA-Qwen2-7B (Perception, Reasoning, Language) | Self-generated 118k caption w/ DD&CoD, 45k VQA w/ DA&CoT, 50k textual QA | | 75.45 | 48.60 | 84.35 | 52.52 | 54.48 | 38.80 | 77.44 | 76.20 |
| **High-Resolution** | | | | | | | | | | |
| *Base Model* | | | | | | | | | | |
| LLaVA-Qwen2-7B-UHD (Guo et al., 2024) | - | | 77.63 | 48.93 | 87.31 | 70.18 | 69.96 | 38.90 | 77.29 | 74.94 |
| *Strong-to-Weak Distillation (Perception)* | | | | | | | | | | |
| LLaVA-NeXT-34B-LLaVA-Qwen2-7B-UHD | 118k Caption w/ DD by LLaVA-NeXT-34B | | 77.75 | 50.60 | 86.46 | 71.20 | 71.56 | 36.90 | 78.38 | 76.00 |
| Qwen2-VL-72B-Instruct-LLaVA-Qwen2-7B-UHD | 118k Caption w/ DD by Qwen2-VL-72B-Instruct | | 77.75 | 51.87 | 86.43 | 71.05 | 72.40 | 38.30 | **79.42** | 76.52 |
| *Multi-Agent Collaboration (Perception)* | | | | | | | | | | |
| Multi-Experts-LLaVA-Qwen2-7B-UHD | 118k Caption w/ DD-FGA by base and expert models | | 77.97 | 49.87 | 86.48 | 71.27 | 71.80 | 37.90 | 77.79 | 76.62 |
| *Self-Improving Cognition (Perception & Reasoning)* | | | | | | | | | | |
| SICOG-LLaVA-Qwen2-7B-UHD (Perception) | Self-generated 118k caption w/ DD&CoD | | **78.08** | 51.60 | 87.03 | 72.42 | 73.04 | 39.50 | 77.34 | 77.59 |
| SICOG-LLaVA-Qwen2-7B-UHD (Perception, Reasoning) | Self-generated 118k caption w/ DD&CoD, 45k VQA w/ DA&CoT | | 77.19 | 50.13 | 87.32 | **73.70** | **73.12** | 39.50 | 79.23 | 77.91 |
| SICOG-LLaVA-Qwen2-7B-UHD (Perception, Reasoning, Language) | Self-generated 118k caption w/ DD&CoD, 45k VQA w/ DA&CoT, 50k textual QA | | 77.80 | **52.47** | **87.84** | 73.05 | 72.24 | **41.40** | 79.42 | **78.40** |
| LLaVA-Llama3.1-8B-UHD | - | | 72.14 | 43.93 | **87.85** | 64.32 | 64.64 | 33.90 | 74.96 | 71.70 |
| SICOG-LLaVA-Llama3.1-8B-UHD (Perception) | Self-generated 118k caption w/ DD&CoD | | 71.92 | **44.80** | 87.37 | 65.09 | 64.96 | 35.70 | 74.52 | 71.11 |
| SICOG-LLaVA-Llama3.1-8B-UHD (Perception, Reasoning) | Self-generated 118k caption w/ DD&CoD, 45k VQA w/ DA&CoT | | 72.03 | 43.20 | 87.38 | **65.78** | **65.56** | 35.90 | **76.15** | **72.05** |
| SICOG-LLaVA-Llama3.1-8B-UHD (Perception, Reasoning, Language) | Self-generated 118k caption w/ DD&CoD, 45k VQA w/ DA&CoT, 50k textual QA | | **72.31** | 43.07 | 87.21 | 64.95 | 65.00 | 33.30 | 75.56 | 70.76 |

Additionally, **SICOG achieves leading performance in fine-grained evaluations of six core capabilities** (Appendix K). **Scaling up self-generated data further enhances SICOG's performance** (Appendix G). **SICOG remains effective when varying recaptioned images** (Appendix H) and **contributes to next-generation foundation MLLMs through continuous cognitive self-improvement**

(Appendix I). These findings collectively demonstrate SICOG's effectiveness in advancing multimodal cognitive abilities in MLLMs.

## 4.3 CAN SICOG FACILITATE A STRONGER REASONING FOUNDATION FOR PROTOTYPING CHAIN-OF-THOUGHT REASONERS DURING POST-TRAINING?

Table 2: Evaluation results of fine-tuning foundation MLLMs to build a CoT reasoner via supervised fine-tuning on 35k CoT reasoning examples (curated in Section 3). P., R., and L. refer to perception, reasoning, and language, respectively.

| Method | Inference | Comprehensive | | | Hallu. | Chart/Table | | Knowledge | | |
|---|---|---|---|---|---|---|---|---|---|---|
| | | MMBen. | MMStar | MMVet | POPE | DocV. | Chart. | Math. | Science. | AI2D |
| *Base Model* | | | | | | | | | | |
| LLaVA-Qwen2-7B-UHD | Direct | **77.63** | 48.93 | 38.26 | **87.31** | 70.18 | 69.96 | 38.90 | 77.29 | **74.94** |
| LLaVA-Qwen2-7B-UHD + Finetune w/ 35k VQA (CoT) | CoT | 72.09 | 49.87 | 41.06 | 85.32 | 69.08 | 77.48 | 44.90 | 84.88 | 72.12 |
| *Self-Improving Cognition* | | | | | | | | | | |
| SICOG-LLaVA-Qwen2-7B-UHD (P., R., L.) + Finetune w/ 35k VQA (CoT) | CoT | 71.97 | **51.00** | **47.29** | 86.34 | **70.76** | 79.24 | **45.70** | **85.77** | 74.09 |

We validate the efficacy of SICOG in strengthening the reasoning foundation for constructing CoT reasoners during post-training. Specifically, we adopt a supervised fine-tuning approach, refining both the base model, LLaVA-Qwen2-7B-UHD, and SICOG-LLaVA-Qwen2-7B-UHD on the 35k CoT reasoning dataset curated in Section 3.

**SICOG establishes a stronger foundation for prototyping a CoT reasoner.** Table 2 demonstrates that post-training the SICOG-LLaVA-Qwen2-7B-UHD outperforms the post-trained baseline across most benchmarks, with up to 6% higher accuracy on MMVet.

**Solely enhancing CoT reasoning may compromise perception abilities.** We observe a significant performance drop on MMBench, which assesses a diverse range of perception tasks. This suggests that prioritizing CoT reasoning in MLLMs can inadvertently impair perception capabilities, underscoring the trade-off between reasoning and perception and the need for balanced optimization. Moreover, we provide **an in-depth analysis (both quantitative and qualitative) of how SICOG enhances the reasoning capabilities of foundation MLLMs** in Appendix F.

## 4.4 CAN PREFERENCE LEARNING SUPPORT SICOG'S SYSTEMATIC PERCEPTION AND REASONING DEVELOPMENT?

Table 3: Evaluation results of different training methods for developing perception and reasoning in LLaVA-Qwen2-7B during Step 1 of SICOG (post-training optimization, Section 3).

| Method | Capability Development | Comprehensive | | | Hallu. | Chart & Table | | Knowledge | | |
|---|---|---|---|---|---|---|---|---|---|---|
| | | MMBen. | MMStar | MMVet | POPE | DocV. | Chart. | Math. | Science. | AI2D |
| *Base Model* | | | | | | | | | | |
| LLaVA-Qwen2-7B | - | 74.44 | 46.67 | 38.85 | 84.55 | 50.62 | 52.72 | 38.00 | 74.91 | 73.77 |
| *Self-Improving Cognition* | | | | | | | | | | |
| SICOG-LLaVA-Qwen2-7B (Per., Rea., Lan.) | SFT (Per., Rea.) | 75.45 | 48.60 | 37.84 | 84.35 | 52.52 | 54.48 | 38.80 | 77.44 | 76.20 |
| | DPO (Per.), SFT (Rea.) | **76.18** | 48.40 | 38.72 | 83.53 | 52.20 | 54.80 | 39.20 | **77.49** | 75.78 |
| | DPO (Per., Rea.) | 74.83 | **49.00** | **38.90** | **84.85** | 52.54 | 55.64 | **41.00** | 76.20 | **76.33** |

We explore the application of preference learning to enhance MLLMs' multimodal perception and reasoning capabilities during Step 1 of SICOG (post-training optimization, Section 3). Specifically, we construct preference caption pairs by selecting high-quality captions from the annotated caption dataset (Section 3) as preferred captions and pairing them with corresponding low-quality (dispreferred) captions. The low-quality captions are generated by corrupting the associated images (details provided in Appendix E). We fine-tune the MLLM on these caption preference pairs using the Direct Preference Optimization (DPO) algorithm (Rafailov et al., 2023) to initialize systematic perception capabilities. Similarly, we extend preference learning to foster systematic reasoning development.

**Preference learning is more effective than supervised fine-tuning for systematic perception and reasoning development.** Preference learning with DPO consistently surpasses standard supervised

fine-tuning across all benchmarks for initializing systematic perception and reasoning in SICOG. For example, on MathVista, preference learning improves accuracy by approximately 2% on the low-resolution model LLaVA-Qwen2-7B, which is particularly challenging to enhance due to inherent visual perception limitations. These results underscore the importance of learning not only from correct examples but also from avoiding mistakes, thereby fostering more robust skill development. We provide a detailed analysis in Appendix E.

### 4.5 HOW DO *Chain-of-Description* AND CHAIN-OF-THOUGHT IMPROVE COGNITION?

Table 4: Evaluation of re-captioning quality comparing the perception-enhanced models fine-tuned on curated caption data in three formats: detailed description (Detailed D), *Chain-of-Description* (CoD), and their combination (Section 3). Metrics (rated 1-5): salient content, fine-grained details, relational attributes, peripheral content, faithfulness, and world knowledge. "Caption": standard format; "Multi.": CoD step-by-step format (see Table 11 in Appendix M for details). Complete results in Appendix J.

| Method | # Avg. Tokens | Systematic Perception | | | | General Performance | |
|---|---|---|---|---|---|---|---|
| | | Sali. | Fine-Grain. | Rela. | Peri. | Faith. | Know. |
| LLaVA-Qwen2-7B-UHD | 135.08 | 4.77 | 4.30 | 3.99 | 3.81 | 4.41 | 3.84 |
| + Finetune w/ Detailed D | 140.73 | 4.71 | 4.52 | 3.92 | 3.91 | 4.20 | 3.77 |
| + Finetune w/ CoD (Caption) | 126.93 | 4.78 | 4.58 | 4.11 | 3.90 | 4.57 | 3.93 |
| + Finetune w/ CoD (Multi.) | 453.13 | 4.82 | 4.80 | 4.74 | 4.29 | 4.57 | 4.01 |
| + Finetune w/ Detailed D & CoD (Detailed D) | 136.50 | 4.76 | 4.67 | 4.01 | 3.82 | 4.51 | 3.88 |
| + Finetune w/ Detailed D & CoD (CoD Multi.) | **453.26** | **4.91** | **4.87** | **4.78** | **4.32** | **4.71** | **4.05** |
| LLaVA-NeXT-34B (Liu et al., 2024b) | 206.50 | 4.77 | 4.51 | 4.04 | 3.95 | 4.59 | 4.12 |

**How Does *Chain-of-Description* Facilitate Multimodal Perception? (Quantitative Analysis.)** We analyze captions for 100 images randomly sampled from BLIP-558k (Li et al., 2022), which is used as unlabeled image captioning data in Section 4. These captions are generated by perception-enhanced models fine-tuned on annotated caption data in three formats: detailed description (Detailed D), *Chain-of-Description* (CoD), and their combination (as implemented in SICOG, described in Section 3). Using GPT-4 with the prompt shown in Table 21, we evaluate six key dimensions. For a holistic analysis, we also include LLaVA-NeXT-34B, a leading open-source MLLM known for its strong captioning capabilities (Li et al., 2024a). Table 9 shows that the base model, regardless of resolution, consistently underperforms in salient content, fine-grained details, relational attributes, and peripheral content. These results highlight the importance of the four-step perception analysis design used in CoD.

***Chain-of-Description* shows strong efficacy in facilitating systematic perception across six key dimensions.** Perception-enhanced models fine-tuned with *Chain-of-Description* outperform those trained on detailed descriptions in both single-step (caption-only) and multi-step formats. Notably, their combination achieves the highest evaluation scores, surpassing LLaVA-NeXT-34B in five of the six dimensions. Furthermore, *Chain-of-Description* generates the longest average caption lengths (approximately 430–450 tokens), indicating a robust perceptual capacity. Additional analysis is provided in Appendix L. Due to space constraints, we provide **the qualitative analysis of *Chain-of-Description* and a detailed discussion of structured Chain-of-Thought** in Appendix J.

## 5 CONCLUSION

We present SICOG, a self-learning framework for constructing next-generation foundation MLLMs by injecting multimodal knowledge and enhancing systematic cognition through multimodal pre-training with self-generated data. Extensive experiments demonstrate that SICOG produces a next-generation MLLM with significantly enhanced cognitive abilities, outperforming existing pre-training approaches across a wide range of benchmarks. Notably, we empirically validate that integrating pre-training with downstream mechanisms—such as post-training optimization and inference-time computation—enables more effective model development, establishing a foundation for a complete self-improving paradigm. For future work, we aim to incorporate embodied experiential data (Zhao et al., 2025) to further enhance real-world application capabilities.

ETHICS STATEMENT

Throughout our research, we have adhered to ethical guidelines that prioritize privacy, fairness, and the well-being of all individuals and groups. All benchmark datasets used in this study are intended solely for research purposes and do not contain any personally identifiable information, thereby safeguarding user privacy. To elicit *Chain-of-Description* data, we carefully designed prompts to avoid language that could be biased or discriminatory toward any individual or group. These measures were implemented to minimize potential negative impacts on users. Furthermore, all self-generated datasets were manually verified to ensure they are free from offensive content, misinformation, and personally identifiable information. To ensure ethical integrity, prompts used for data generation were carefully designed to exclude biased or discriminatory language. All generated data was manually reviewed to confirm it contains no offensive material, misinformation, or personally identifiable information.

REPRODUCIBILITY STATEMENT

We provide all necessary implementation details in Section 4 and Appendix M. The source code and raw data are included in the supplementary materials, along with detailed instructions in the `README.md` file. All results are easily reproducible.

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

CONTENTS

## A    THE USE OF LARGE LANGUAGE MODELS (LLMS)

In this paper, we utilized LLMs, specifically GPT-4o (Hurst et al., 2024), for two limited purposes: (1) to evaluate the quality of generated captions using our specially designed rubrics (detailed in Appendix J); and (2) to assist in refining the manuscript's language for clarity and fluency. LLMs were not involved in research ideation, experimental design, or drafting the initial version of the paper.

# B    THE COMPREHENSIVE ILLUSTRATION OF SICOG

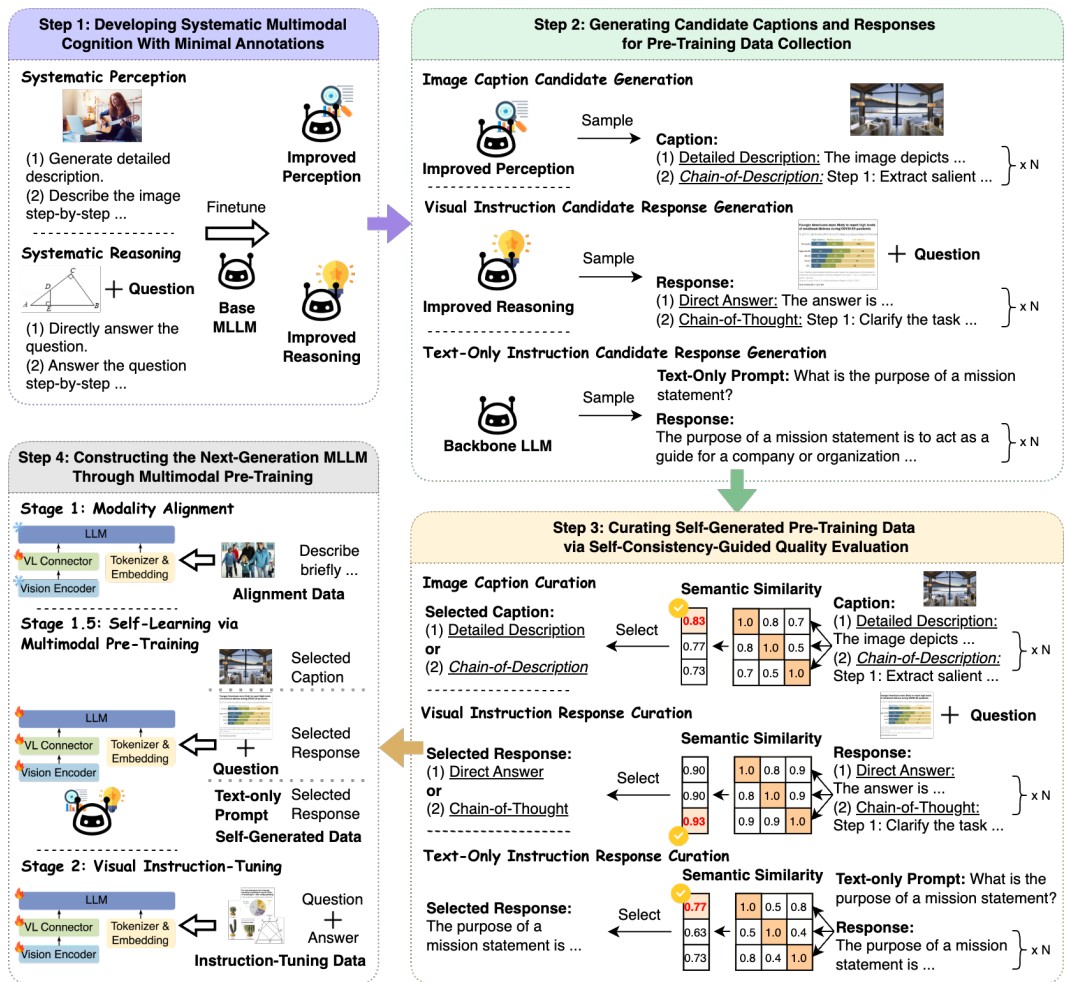

Figure 3: The SICOG framework consists of four steps: (*i*) **Enhancing multimodal cognition:** Fine-tune an MLLM using minimal annotated data—image-captioning data in the *Chain-of-Description* format and visual instruction-tuning data with structured CoT—to improve systematic perception and reasoning (upper left). (*ii*) **Generating candidate data:** Use the improved models to sample candidate captions and responses for pre-training (upper right). (*iii*) **Curating pre-training data:** Apply self-consistency-guided quality evaluation to select the most semantically consistent, self-generated candidates for learning (lower right). (*iv*) **Constructing the next-generation MLLM:** Perform multimodal pre-training on the curated data to build a foundation MLLM with enhanced self-improving cognition (lower left).

The goal of SICOG is to advance MLLMs by equipping them with rich multimodal knowledge and systematic cognitive capabilities—namely, systematic visual understanding ("how to observe") and in-depth multimodal reasoning ("how to think")—during multimodal pre-training, with minimal reliance on external annotations. As illustrated in Figure 2, SICOG operates through four key steps.

**Step 1: Developing systematic multimodal cognitive capabilities with minimal annotated data.** We first equip a given MLLM $\mathcal{M}$, parameterized by $\theta$, with systematic perception and reasoning abilities while using *minimal* annotated multimodal data. This involves fine-tuning the model to develop structured, multi-step perception and reasoning chains, enabling it to systematically process and integrate multimodal information.

Specifically, this step consists of the following two core components:

- **Systematic multimodal perception.** To enhance the MLLM's ability to systematically observe and interpret images, we fine-tune $\mathcal{M}$ using a combination of image-captioning datasets, yielding a model with improved perception, $\mathcal{M}_0^{\text{Perception}}$. Specifically, these datasets consist of images $v$, prompts $x$, intermediate step-by-step analyses $s$, and descriptions $y$, structured in two formats of captions: Detailed Description (DD) and *Chain-of-Description* (CoD) (see Section 3.2 for details on the *Chain-of-Description* strategy and data collection.)

$$\mathcal{D}^{\text{Perception}} = \mathcal{D}_{DD}^{\text{Perception}} + \mathcal{D}_{CoD}^{\text{Perception}} = \{(v_i, x_i, y_i)\}_{i=1}^N + \{(v_i, x_i, s_i, y_i)\}_{i=1}^N, \quad (6)$$

where $N$ is the number of samples in each dataset. The model is fine-tuned using the following objective, improving its multimodal perception:

$$\mathcal{M}_0^{\text{Perception}} \leftarrow \mathcal{J}_\theta(\mathcal{D}^{\text{Perception}}) = \sum_{i=1}^N [\log p_\theta(y \mid v, x) + \log p_\theta(s, y \mid v, x)] \quad (7)$$

- **Systematic multimodal reasoning.** Similarly, to strengthen the model's systematic and in-depth reasoning capabilities, we fine-tune $\mathcal{M}$ using a mix of visual instruction-tuning datasets, yielding a model with enhanced reasoning, $\mathcal{M}_0^{\text{Reasoning}}$. These datasets consist of images $v$, questions $q$, intermediate step-by-step rationales $r$, and answers $a$, structured in two formats of responses: *Direct Answer* (DA) and *Chain-of-Thought* (CoT) (see Section 3.3 for details on data curation).

$$\mathcal{D}^{\text{Reasoning}} = \mathcal{D}_{DA}^{\text{Reasoning}} + \mathcal{D}_{CoT}^{\text{Reasoning}} = \{(v_i, q_i, a_i)\}_{i=1}^M + \{(v_i, q_i, r_i, a_i)\}_{i=1}^M, \quad (8)$$

where $M$ is the number of samples in each dataset. The model is fine-tuned using the following objective, fostering its multimodal reasoning:

$$\mathcal{M}_0^{\text{Reasoning}} \leftarrow \mathcal{J}_\theta(\mathcal{D}^{\text{Reasoning}}) = \sum_{i=1}^M [\log p_\theta(a \mid v, q) + \log p_\theta(r, a \mid v, q)] \quad (9)$$

**Step 2: Generating candidate captions and responses for pre-training data collection.** Next, we construct multimodal pre-training data by leveraging the improved models, $\mathcal{M}_0^{\text{Perception}}$ and $\mathcal{M}_0^{\text{Reasoning}}$, to generate candidate image captions and visual instruction responses. Additionally, to mitigate potential degradation of the MLLM's language capabilities during multimodal pre-training, we prompt the backbone large language model (LLM), $\mathcal{M}_{LLM}$, to generate candidate responses for text-only instructions.

This step consists of three key components:

- **Image caption candidate generation.** Given a collection of unlabeled images $\{v_k\}_{k=1}^K$, we prompt $\mathcal{M}_0^{\text{Perception}}$ (with policy $p_{\mathcal{M}_0^{\text{Perception}}}$) using two types of prompts to generate detailed descriptions and induce *Chain-of-Description* perception:

  1. "Please generate a detailed caption of this image. Be as descriptive as possible." ($x_{DD}$).
  2. "Please generate a detailed caption of this image. Describe the image step by step." ($x_{CoD}$).

  Specifically, for a given image $v_k$, the model $\mathcal{M}_0^{\text{Perception}}$ generates multiple candidate captions via sampling:

$$\{\hat{y}_k\} \sim p_{\mathcal{M}_0^{\text{Perception}}}(\cdot \mid v_k, x_{DD}),$$
$$\{(\hat{s}_k, \hat{y}_k)\} \sim p_{\mathcal{M}_0^{\text{Perception}}}(\cdot \mid v_k, x_{CoD}), \quad (10)$$

  where $\{\hat{y}_k\}$ represents the set of detailed descriptions, and $\{(\hat{s}_k, \hat{y}_k)\}$ represents the set of step-by-step analyses along with corresponding detailed descriptions. This results in a collection of candidate image captions in two formats:

$$\mathcal{D}_{\text{Cand}}^{\text{Perception}} = \{(v_k, x_{DD}, \{\hat{y}_k\})\}_{k=1}^K + \{(v_k, x_{CoD}, \{(\hat{s}_k, \hat{y}_k)\})\}_{k=1}^K. \quad (11)$$

- **Visual instruction candidate response generation.** Similarly, given a collection of unlabeled images $\{v_z\}_{z=1}^{Z}$ with corresponding questions $q_z$, we prompt $\mathcal{M}_0^{\text{Reasoning}}$ (with policy $p_{\mathcal{M}_0^{\text{Reasoning}}}$) using two types of prompts to generate direct answers (DA) and induce *Chain-of-Thought* (CoT) reasoning:

  1. "`<original question>.`" ($q_{DA}$).
  2. "`<original question> Answer the question step by step.`" ($q_{CoT}$).

  Specifically, for a given image $v_z$ and corresponding question $q_z$, the model $\mathcal{M}_0^{\text{Reasoning}}$ generates multiple candidate responses:

  $$\begin{aligned} \{\hat{a}_z\} &\sim p_{\mathcal{M}_0^{\text{Reasoning}}}(\cdot \mid v_z, q_{DA}), \\ \{(\hat{r}_z, \hat{a}_z)\} &\sim p_{\mathcal{M}_0^{\text{Reasoning}}}(\cdot \mid v_z, q_{CoT}), \end{aligned} \quad (12)$$

  where $\{\hat{a}_z\}$ represents the set of direct answers, and $\{(\hat{r}_z, \hat{a}_z)\}$ represents the set of step-by-step rationales along with corresponding answers. This results in a collection of candidate visual instruction responses in two formats:

  $$\mathcal{D}_{\text{Cand}}^{\text{Reasoning}} = \{(v_z, q_{DA}, \{\hat{a}_z\})\}_{z=1}^{Z} + \{(v_z, q_{CoT}, \{(\hat{r}_z, \hat{a}_z)\})\}_{z=1}^{Z}. \quad (13)$$

- **Text-only instruction candidate response generation.** To maintain language capabilities, we generate textual instruction responses using the backbone LLM, $\mathcal{M}_{LLM}$ (with policy $p_{\mathcal{M}_{LLM}}$), based on a collection of unlabeled text prompts $\{x_t\}_{t=1}^{T}$.

  Specifically, for a given prompt $x_t$, the model $\mathcal{M}_{LLM}$ generates a set of candidate responses:

  $$\{\hat{y}_t\} \sim p_{\mathcal{M}_{LLM}}(\cdot \mid x_t), \quad (14)$$

  resulting in a collection of candidate textual instruction responses:

  $$\mathcal{D}_{\text{Cand}}^{\text{Language}} = \{(x_t, \{\hat{y}_t\})\}_{t=1}^{T}. \quad (15)$$

**Step 3: Curating self-generated pre-training data via self-consistency-guided quality evaluation.** To ensure the quality of self-generated pre-training data, we employ *self-consistency* to evaluate candidate samples without external supervision. This method is based on the principle that higher-quality candidates exhibit greater semantic consistency (Wu et al., 2024). The most consistent candidates are selected for further self-refinement during multimodal pre-training.

Specifically, we apply a semantic-similarity-guided self-consistency evaluation function, $f(\cdot)$. For each instance (*e.g.,* an unlabeled image), it assesses the quality of candidates (*e.g.,* candidate captions) by comparing each candidate against all others based on semantic similarity and selects the candidate with the highest consistency score, provided it exceeds a predefined threshold $\tau$ (*i.e.,* otherwise, the instance and its candidates are skipped):

$$f(\{c\}) = \arg\max_{c \in \{c\}} \frac{1}{N_{\text{cand}}} \sum_{j=1}^{N_{\text{cand}}} \text{sim}(c, c^{(j)}), \quad s.t. \quad \max_{c \in \{c\}} \frac{1}{N_{\text{cand}}} \sum_{j=1}^{N_{\text{cand}}} \text{sim}(c, c^{(j)}) \geq \tau, \quad (16)$$

where $N_{\text{cand}}$ is the number of candidate samples being compared, and $\{c\}$ is the candidate set.

We apply this method as follows:

- **Self-generated image caption curation.** For each image $v_k$, we apply $f(\cdot)$ to assess the quality of candidate captions by comparing each generated description in the caption against all others. The most consistent caption is selected as the final self-generated caption:

  $$\hat{y}_{\text{selected}} \vee (\hat{s}_{\text{selected}}, \hat{y}_{\text{selected}}) = f(\{\hat{y}_k\} \cup \{(\hat{s}_k, \hat{y}_k)\})$$

  $$= \arg\max_{y \in \{\hat{y}_k\} \cup \{(\hat{s}_k, \hat{y}_k)\}} \frac{1}{N_{\text{cand}}} \sum_{j=1}^{N_{\text{cand}}} \text{sim}(y, y^{(j)}),$$

  $$s.t. \max_{y \in \{\hat{y}_k\} \cup \{(\hat{s}_k, \hat{y}_k)\}} \frac{1}{N_{\text{cand}}} \sum_{j=1}^{N_{\text{cand}}} \text{sim}(y, y^{(j)}) \geq \tau^{\text{Perception}}.$$

  $$(17)$$

where $N_{\text{cand}} = |\{\hat{y}_k\} \cup \{(\hat{s}_k, \hat{y}_k)\}|$ is the total number of candidate captions for each image. The curated dataset of self-generated image captions is:

$$\mathcal{D}_{\text{Selected}}^{\text{Perception}} = \{(v, x_{DD}, \hat{y}_{\text{selected}}) \vee (v, x_{CoD}, \hat{s}_{\text{selected}}, \hat{y}_{\text{selected}})\}_{k=1}^{K}. \qquad (18)$$

- **Self-generated visual instruction response curation.** Similarly, for each image $v_z$ and corresponding question $q_z$, we apply $f(\cdot)$ to evaluate candidate responses, selecting the most consistent response:

$$\hat{a}_{\text{selected}} \vee (\hat{r}_{\text{selected}}, \hat{a}_{\text{selected}}) = f(\{\hat{a}_z\} \cup \{(\hat{r}_z, \hat{a}_z)\})$$

$$= \arg \max_{a \in \{\hat{a}_z\} \cup \{(\hat{r}_z, \hat{a}_z)\}} \frac{1}{N_{\text{cand}}} \sum_{j=1}^{N_{\text{cand}}} \text{sim}(a, a^{(j)}),$$

$$s.t. \quad \max_{a \in \{\hat{a}_z\} \cup \{(\hat{r}_z, \hat{a}_z)\}} \frac{1}{N_{\text{cand}}} \sum_{j=1}^{N_{\text{cand}}} \text{sim}(a, a^{(j)}) \geq \tau^{\text{Reasoning}}.$$

$$\qquad (19)$$

where $N_{\text{cand}} = |\{\hat{a}_z\} \cup \{(\hat{r}_z, \hat{a}_z)\}|$ is the total number of candidate responses for each question. The curated set of self-generated visual instruction responses is:

$$\mathcal{D}_{\text{Selected}}^{\text{Reasoning}} = \{(v_z, q_{DA}, \hat{a}_{\text{selected}}) \vee (v_z, q_{CoT}, \hat{r}_{\text{selected}}, \hat{a}_{\text{selected}})\}_{z=1}^{Z}. \qquad (20)$$

- **Self-generated text-only instruction response curation.** Similarly, for each prompt $x_t$, we apply $f(\cdot)$ to evaluate candidate responses, selecting the most consistent response:

$$\hat{y}_{\text{t-selected}} = f(\{\hat{y}_t\})$$

$$= \arg \max_{y_t \in \{\hat{y}_t\}} \frac{1}{N_{\text{cand}}} \sum_{j=1}^{N_{\text{cand}}} \text{sim}(y_t, y_t^{(j)})$$

$$s.t. \quad \max_{y_t \in \{\hat{y}_t\}} \frac{1}{N_{\text{cand}}} \sum_{j=1}^{N_{\text{cand}}} \text{sim}(y_t, y_t^{(j)}) \geq \tau^{\text{Language}}.$$

$$\qquad (21)$$

where $N_{\text{cand}} = |\{\hat{y}_t\}|$ is the total number of candidate responses for each question. The curated set of self-generated textual instruction responses is:

$$\mathcal{D}_{\text{Selected}}^{\text{Language}} = \{(x_t, \hat{y}_{\text{t-selected}})\}_{t=1}^{T}. \qquad (22)$$

Finally, we obtain the curated self-generated multimodal pre-training dataset:

$$\mathcal{D}^{\text{Pre-training}} = \mathcal{D}_{\text{Selected}}^{\text{Perception}} + \mathcal{D}_{\text{Selected}}^{\text{Reasoning}} + \mathcal{D}_{\text{Selected}}^{\text{Language}}. \qquad (23)$$

**Step 4: Constructing the next-generation MLLM through multimodal pre-training.** To build the next-generation foundation MLLM, we introduce an intermediate multimodal pre-training stage, Stage 1.5, within the standard two-stage training strategy, following Liu et al. (2024b); Li et al. (2024a). This stage refines the MLLM using curated self-generated pre-training data. The complete training strategy consists of three stages, as shown in the lower left part in Figure 2:

- **Stage 1: Modality alignment.** In this stage, image features are aligned with the text embedding space. Following Li et al. (2024a), we train only the vision-language (VL) connector using image-text pairs from $\mathcal{D}^{\text{Alignment}}$, while keeping the vision encoder (*e.g.,* vision transformer) and large language model (LLM) frozen.
- **Stage 1.5: Self-learning via multimodal pre-training.** The model undergoes training with curated self-generated pre-training data $\mathcal{D}^{\text{Pre-Training}}$ to acquire multimodal knowledge from these self-generated samples and internalize its systematic multimodal perception and reasoning abilities. During this stage, all model components are fully trainable.

- **Stage 2: Visual instruction-tuning.** In the final stage, the model is fine-tuned using instruction-tuning data $\mathcal{D}^{\text{Instruction-Tuning}}$ to develop robust visual instruction-following capabilities, with all model components fully trainable.

This three-stage training process is formulated as follows, resulting in the next-generation foundation MLLM with self-improved cognition $\mathcal{M}^{\text{Next}}$:

$$
\begin{aligned}
\mathcal{M}^{1} &\leftarrow \mathcal{L}_{\phi}^{\text{Stage 1}}(\mathcal{D}^{\text{Alignment}}) \\
\mathcal{M}^{1.5} &\leftarrow \mathcal{L}_{\phi}^{\text{Stage 1.5}}(\mathcal{D}^{\text{Pre-training}}) \\
\mathcal{M}^{\text{Next}} &\leftarrow \mathcal{L}_{\phi}^{\text{Stage 2}}(\mathcal{D}^{\text{Instruction-Tuning}}).
\end{aligned}
\tag{24}
$$

## B.1 ENHANCING SYSTEMATIC PERCEPTION WITH *Chain-of-Description*

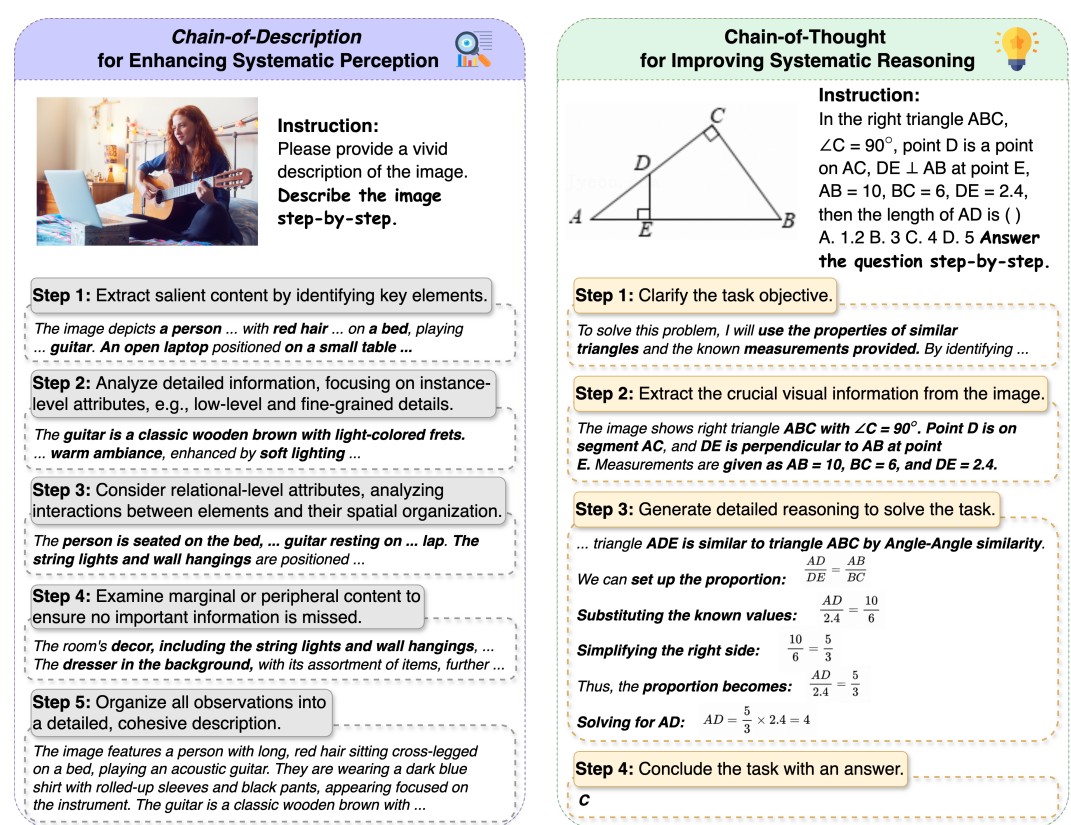

Figure 4: Illustration of *Chain-of-Description* (left) for enhancing systematic perception and structured Chain-of-Thought (right) for strengthening reasoning capabilities.

We introduce *Chain-of-Description* (CoD) to enable systematic and comprehensive perception, equipping the MLLM with the ability to logically analyze and describe visual information step by step ("how to observe"). This structured approach enhances the MLLM's efficiency in thoroughly interpreting visual content. Specifically, *Chain-of-Description* perception is organized into the following five steps (Figure 4, left):

- **Step 1: Extract salient content.** Identify the key elements that define the overall context and meaning of the image, laying the foundation for basic visual recognition.

- **Step 2: Analyze detailed information.** Focus on instance-level attributes, such as low-level and fine-grained details, *e.g.,* "the guitar is a classic wooden brown with light-colored frets." This step ensures a precise and detailed interpretation of the image.

- **Step 3: Consider relational-level attributes.** Analyze interactions between elements and their spatial organization, *e.g.,* "the person is seated on the bed," leading to a richer and more comprehensive understanding of visual relationships.

- **Step 4: Examine marginal or peripheral content.** Pay attention to less prominent or background details, *e.g.,* "the dresser in the background," to ensure no important information is overlooked.

- **Step 5: Organize all observations.** Synthesize all findings into a cohesive, detailed description, enabling comprehensive coverage and holistic image understanding.

**Data preparation.** To enable systematic perception in MLLMs, we utilize GPT-4o (Hurst et al., 2024) with manually curated prompts (Table 20) to generate detailed, step-by-step analyses of visual features. Specifically, we prompt GPT-4o to recaption 35k images from the Vision-Flan dataset (Xu et al., 2024b), which provides diverse visual content. A detailed example is presented in Table 22.

## B.2 IMPROVING SYSTEMATIC REASONING WITH STRUCTURED CHAIN-OF-THOUGHT

We adopt a structured Chain-of-Thought (CoT) approach (Xu et al., 2025) to enhance systematic and in-depth reasoning. For completeness, we briefly summarize this approach. It enables the MLLM to decompose problem-solving into logical steps: analyzing multimodal questions, gathering relevant visual information, and answering progressively. Specifically, the structured CoT process (Figure 4, right) follows four logical steps: (*i*) **Step 1: Clarify the task objective.** Identify the problem's requirements and constraints, establishing a foundational understanding. (*ii*) **Step 2: Extract crucial visual information.** Identify and extract relevant visual elements to enhance multimodal comprehension. (*iii*) **Step 3: Generate detailed reasoning.** Construct a logical sequence of intermediate steps based on the extracted information to derive an answer systematically. (*iv*) **Step 4: Conclude with an answer.** Synthesize the reasoning steps into a coherent and accurate final response.

**Data preparation.** To enable systematic reasoning in the MLLM, we revise the dataset curated by (Xu et al., 2025). Specifically, we randomly select 35k training examples from `LLaVA-CoT`, covering ten well-studied QA tasks, and replace the special tags (*e.g.,* `<SUMMARY>` and `</SUMMARY>`) with curated step-by-step instructions. A detailed example is shown in Table 23.

The complete training process is summarized in Algorithm 1, with implementation details in Section 4.1 and Appendix M.

---

**Algorithm 1 SICOG: A Self-Learning Framework for Systematic Multimodal Cognition**

---

1: **Input:** Pretrained MLLM $\mathcal{M}$, parameterized by $\theta$

2:     Systematic perception data: $\mathcal{D}^{\text{Perception}} = \mathcal{D}^{\text{Perception}}_{DD} + \mathcal{D}^{\text{Perception}}_{CoD}$

3:     Systematic reasoning data: $\mathcal{D}^{\text{Reasoning}} = \mathcal{D}^{\text{Reasoning}}_{DA} + \mathcal{D}^{\text{Reasoning}}_{CoT}$

4:     Default alignment data: $\mathcal{D}^{\text{Alignment}}$, instruction-tuning data: $\mathcal{D}^{\text{Instruction-Tuning}}$

5:     Unlabeled data: (1) unlabeled image sets $\{v_k\}$ with prompts $x$, (2) unlabeled image sets $\{v_z\}$ with questions $q$, (3) unlabeled text-only prompts $x_t$

6: **Goal:** Enable systematic visual understanding and reasoning via self-learning

7: $\mathcal{M}_0 \leftarrow \mathcal{M}$                                  # Initialize model

8: **for** $n = 1, \ldots, N$ **do**           # Iterative foundation MLLM update, when applicable

9:     **Step 1: Systematic Multimodal Cognitive Training**

10:       Fine-tune perception, reasoning models:

$$\mathcal{M}^{\text{Perception}}_{n-1} \leftarrow \mathcal{J}_\theta(\mathcal{D}^{\text{Perception}}), \mathcal{M}^{\text{Reasoning}}_{n-1} \leftarrow \mathcal{J}_\theta(\mathcal{D}^{\text{Reasoning}})$$

11:     **Step 2: Generating Candidate Captions and Responses**

12:       Generate image captions:                # Foster multimodal perception

$$\{\hat{y}_k\}, \{(\hat{s}_k, \hat{y}_k)\} \sim p_{\mathcal{M}^{\text{Perception}}_{n-1}}(\cdot \mid v_k, x)$$

13:       Generate visual instruction responses:      # Enhance multimodal reasoning

$$\{\hat{a}_z\}, \{(\hat{r}_z, \hat{a}_z)\} \sim p_{\mathcal{M}^{\text{Reasoning}}_{n-1}}(\cdot \mid v_z, q)$$

14:       Generate text-only responses:               # Maintain language

$$\{\hat{y}_t\} \sim p_{\mathcal{M}_{LLM}}(\cdot \mid x_t)$$

15:     **Step 3: Self-Consistency Selection**

16:       Select the optimal candidates based on self-consistency, using the predefined threshold $\tau$:

$$\mathcal{D}^{\text{Perception}}_{\text{Selected}} \leftarrow \arg\max_y \sum_j \text{sim}(y, y^{(j)}) \quad s.t. \quad \max_y \frac{1}{j} \sum_j \text{sim}(y, y^{(j)}) \geq \tau^{\text{Perception}}$$

$$\mathcal{D}^{\text{Reasoning}}_{\text{Selected}} \leftarrow \arg\max_a \sum_j \text{sim}(a, a^{(j)}) \quad s.t. \quad \max_a \frac{1}{j} \sum_j \text{sim}(a, a^{(j)}) \geq \tau^{\text{Reasoning}}$$

$$\mathcal{D}^{\text{Language}}_{\text{Selected}} \leftarrow \arg\max_{y_t} \sum_j \text{sim}(y_t, y_t^{(j)}) \quad s.t. \quad \max_{y_t} \frac{1}{j} \sum_j \text{sim}(y_t, y_t^{(j)}) \geq \tau^{\text{Language}}$$

17:       Construct refined pre-training dataset:

$$\mathcal{D}^{\text{Pre-training}} = \mathcal{D}^{\text{Perception}}_{\text{Selected}} + \mathcal{D}^{\text{Reasoning}}_{\text{Selected}} + \mathcal{D}^{\text{Language}}_{\text{Selected}}$$

18:     **Step 4: Constructing the Next-Generation Foundation MLLM**

19:       **Stage 1: Modality Alignment**

$$\mathcal{M}^1_n \leftarrow \mathcal{L}^{\text{Stage 1}}_\phi(\mathcal{D}^{\text{Alignment}})$$

20:       **Stage 1.5: Multimodal Pre-Training**      # Pre-train on curated self-generated data

$$\mathcal{M}^{1.5}_n \leftarrow \mathcal{L}^{\text{Stage 1.5}}_\phi(\mathcal{D}^{\text{Pre-training}})$$

21:       **Stage 2: Visual Instruction-Tuning**

$$\mathcal{M}^{\text{Next}}_n \leftarrow \mathcal{L}^{\text{Stage 2}}_\phi(\mathcal{D}^{\text{Instruction-Tuning}})$$

22: **end for**

23: **Output:** Next-generation foundation MLLM with self-improved cognition $\mathcal{M}^{\text{Next}}_n$

---

# C  MATHEMATICAL PROOF OF SICOG

## C.1  DEFINITIONS

To formalize the proof for this self-improving cycle, we establish the following definitions:

- Let $M_t$ be the base pre-trained model at the start of iteration $t$.
- Let $D_{\text{pre},t}$ be the large-scale pre-training dataset used to train $M_t$ from scratch.
- Let $D_A$ be a separate, high-quality annotated dataset used for capability enhancement via fine-tuning.
- Let $M_t'$ be the enhanced model created by fine-tuning $M_t$ on the dataset $D_A$. This model is used for data generation but is discarded at the end of the iteration.
- Let $D_{S,t}$ be the new dataset curated at iteration $t$ by applying a quality filter to the outputs of the **enhanced model** $M_t'$.
- Let $D_{\text{pre},t+1}$ be the augmented pre-training dataset for the next iteration: $D_{\text{pre},t+1} = D_{\text{pre},t} \cup D_{S,t}$.
- Let $M_{t+1}$ be the new base pre-trained model for the next iteration, produced by training a model **from scratch** on the complete augmented dataset $D_{\text{pre},t+1}$.
- Let $J(M)$ be the **capability** of a model $M$, defined as its expected true utility on a representative test distribution.
- Let $U(D)$ be the average true utility of the examples within a dataset $D$.

## C.2  THE ITERATIVE CYCLE

At each iteration $t$, the framework executes the following steps:

1. **Capability Enhancement (Post-Training Optimization):** The current base model $M_t$ is fine-tuned on the high-quality dataset $D_A$ to produce an enhanced model, $M_t'$.

2. **Data Generation and Curation (Inference-Time Computation):** The enhanced model $M_t'$ is used to generate a large corpus of outputs. These are filtered to create a high-quality curated dataset, $D_{S,t}$.

3. **Corpus Augmentation:** The curated dataset $D_{S,t}$ is added to the previous pre-training corpus $D_{\text{pre},t}$ to form a new, larger corpus $D_{\text{pre},t+1}$.

4. **Re-training from Scratch:** A new base model, $M_{t+1}$, is initialized with random weights and trained from scratch on the entire augmented dataset $D_{\text{pre},t+1}$.

The objective remains to prove that this process ensures $J(M_{t+1}) \geq J(M_t)$.

## C.3  CORE ASSUMPTIONS

The proof now relies on three fundamental assumptions.

**Assumption 1** (Beneficial Fine-Tuning). Fine-tuning the base model $M_t$ on the high-quality annotated dataset $D_A$ results in an enhanced model $M_t'$ with capability that is greater than or equal to the original model. This captures the benefit of the fine-tuning step.

$$J(M_t') \geq J(M_t) \tag{25}$$

**Assumption 2** (Effective Curation). The curation process is effective. It selects a subset of self-generated data, $D_{S,t}$, whose average utility is strictly greater than the capability of the model that generated it, which is now the **enhanced model** $M_t'$.

$$U(D_{S,t}) > J(M_t') \tag{26}$$

**Assumption 3** (Monotonic Pre-training on Augmented Data). This assumption remains the same but is crucial. It posits that training a model from scratch on an augmented dataset results in a better (or

equal) base model, provided the added data is of sufficiently high quality relative to the **original base model** $M_t$.

$$\text{If } U(D_{S,t}) > J(M_t), \text{ then } J(M_{t+1}) \geq J(M_t) \tag{27}$$

This assumption connects the quality of the new data to the improvement of the next-generation *base model*.

## C.4 THE PROOF OF NON-DECREASING CAPABILITY

We will now prove that $J(M_{t+1}) \geq J(M_t)$ by following the steps of the iterative self-improving cycle.

1. The process begins at iteration $t$ with the base model $M_t$, which has capability $J(M_t)$.

2. The model $M_t$ is fine-tuned on $D_A$ to create the enhanced model $M_t'$. By **Assumption 1 (Beneficial Fine-Tuning)**, we have:

$$J(M_t') \geq J(M_t) \tag{28}$$

3. The enhanced model $M_t'$ is used to generate and curate the new dataset, $D_{S,t}$. By **Assumption 2 (Effective Curation)**, the quality of this new data is superior to the capability of the model that generated it:

$$U(D_{S,t}) > J(M_t') \tag{29}$$

4. We can now combine the inequalities from steps 2 and 3. From equation 29 and equation 28, we can form a logical chain:

$$U(D_{S,t}) > J(M_t') \geq J(M_t) \tag{30}$$

This chain implies that the utility of the new data is strictly greater than the capability of the *original base model*:

$$U(D_{S,t}) > J(M_t) \tag{31}$$

5. A new base model, $M_{t+1}$, is then trained from scratch on the augmented pre-training corpus, $D_{\text{pre},t+1} = D_{\text{pre},t} \cup D_{S,t}$.

6. We are now in the exact scenario described by **Assumption 3 (Monotonic Pre-training on Augmented Data)**. We have:

   - An original base model $M_t$.
   - A new base model $M_{t+1}$ trained on $D_{\text{pre},t} \cup D_{S,t}$.
   - A guarantee from inequality equation 31 that the quality condition $U(D_{S,t}) > J(M_t)$ is met.

7. Therefore, by directly applying **Assumption 3**, we can conclude that the capability of the new base model is greater than or equal to the capability of the original base model:

$$J(M_{t+1}) \geq J(M_t) \tag{32}$$

We have formally demonstrated that even with the intermediate fine-tuning step, the iterative self-improving cycle ensures a monotonically non-decreasing sequence of **base model capabilities**. This holds true as long as our three core assumptions are valid. The dataset $D_A$ now plays a critical role as an "enhancer" within the loop, helping to generate even higher-quality data ($D_{S,t}$) than the base model could on its own, thereby driving the improvement of the entire system.

# D    DETAILED DISCUSSION ON RELATED WORK

Table 5: Comparison of multimodal (vision-language) pre-training methods for enhancing multimodal capabilities. For VILA[2], ✓(✗) indicates a hybrid approach combining bootstrapped captions with fine-grained attributes from expert models. Detailed D (Detailed Description), DD-FGA (Detailed Description with Fine-Grained Attributes, Direct A (Direct Answer).

| Method | w/o External Annotation | Caption Type | | | VQA Type | |
|---|---|---|---|---|---|---|
| | | Detailed D | DD-FGA | CoD | Direct A | CoT |
| *Detailed (Re-)Captioning (Perception)* | | | | | | |
| ALLaVA (Chen et al., 2024a) | ✗ | ✓ | | | | |
| LLaVA-NeXT (Li et al., 2024a) | | | | | | |
| DCE (Sun et al., 2024) | ✗ | | ✓ | | | |
| MMGiC (Xu et al., 2024a) | | | | | | |
| VILA[2] (Fang et al., 2024) | ✓(✗) | | ✓ | | | |
| *Detailed Re-Captioning & Visual Instruction Tuning (Perception & Reasoning)* | | | | | | |
| SICOG (Ours) | ✓ | ✓ | | ✓ | ✓ | ✓ |

**Improving multimodal perception abilities of MLLMs.**    Although MLLMs demonstrate strong multimodal perception capabilities (Liu et al., 2024a; Lu et al., 2024a), they often struggle with fine-grained tasks such as OCR (Fu et al., 2024b; Liu et al., 2024d; Yin et al., 2024; Lai et al., 2023; Li et al., 2024d; Peng et al., 2023). These challenges arise primarily from the reliance on popular large-scale caption datasets (*i.e.,* image-text pairs) (Sharma et al., 2018; Schuhmann et al., 2022; Changpinyo et al., 2021) for modality alignment, which often contain short, coarse-grained captions, restricting their ability to extract detailed visual information (Chen et al., 2024b;a; Lai et al., 2024). One common solution is additional pre-training with high-quality, detailed captions (Chen et al., 2024a; Li et al., 2024d; Lu et al., 2024a; Bai et al., 2023; Yu et al., 2024a) or captions enriched with fine-grained attributes (Xu et al., 2024a; Sun et al., 2024; Fang et al., 2024), improving their ability to capture visual details. In contrast, we propose *Chain-of-Description*, which explicitly models the perception process. This approach trains models to systematically acquire and interpret visual information through step-by-step analysis and decomposition of complex scenes, enabling deeper understanding of fine-grained details.

**Improving multimodal reasoning abilities of MLLMs.**    Complex multimodal reasoning tasks that require integrating visual information into reasoning processes, such as mathematical computation, present significant challenges for MLLMs (Yue et al., 2024b; Chen et al., 2024d; Hao et al., 2025; Xu et al., 2025). Recent studies (Chen et al., 2024d; Cheng et al., 2024; Zhang et al., 2024c) enhance reasoning capabilities by incorporating chain-of-thought (CoT) reasoning (Wei et al., 2022; Zhang et al., 2023), prompting or fine-tuning models to generate intermediate reasoning steps before producing final answers. Structured and systematic extensions of CoT (Xiang et al., 2024; Xu et al., 2025; Cheng et al., 2024; Dong et al., 2024) further improve performance through step-by-step logical processes. While these approaches prove effective during post-training, we investigate incorporating CoT reasoning during pre-training, recognizing this stage as foundational to MLLMs' overall capabilities.

# E  CAN PREFERENCE LEARNING SUPPORT SICOG'S SYSTEMATIC PERCEPTION AND REASONING DEVELOPMENT?

Motivated by the great success of preference learning in adapting MLLMs to follow instructions during the post-training stage (Rafailov et al., 2023; Zhang et al., 2024a), we explore its application to enhance MLLM's multimodal perception and reasoning capabilities during Step 1 of SICOG (Section 3). Specifically, we construct preference caption pairs by using high-quality captions from the annotated caption dataset (Section 3) as preferred captions and pairing them with corresponding low-quality (dispreferred) captions. The low-quality captions are generated by corrupting the associated images through the following methods (Figure 5): (*i*) introducing random noise to hinder key information capture, (*ii*) altering object colors to disrupt fine-grained detail perception, (*iii*) mirroring and rotating images to distort relation-level attributes, and (*iv*) masking peripheral objects to obscure peripheral content. We fine-tune the MLLM on these caption preference pairs using the Direct Preference Optimization (DPO) algorithm (Rafailov et al., 2023) to initialize systematic perception capabilities. Similarly, we extend preference learning to develop systematic reasoning capabilities.

Figure 5: Illustration of (a) the original image and (b) the four types of image corruption.

**Preference learning supports SICOG's systematic perception and reasoning development.** As shown in Table 6, preference learning with DPO significantly enhances MLLMs' systematic perception and reasoning, enabling their self-improvement via SICOG, *e.g.*, achieving a 2.5% accuracy gain on MMstar.

Table 6: Evaluation results of different training methods for developing perception and reasoning in LLaVA-Qwen2-7B during Step 1 of SICOG (post-training optimization, Section 3).

| Method | Capability Development | Comprehensive | | | Hallu. | Chart & Table | | Knowledge | | |
|---|---|---|---|---|---|---|---|---|---|---|
| | | MMBen. | MMStar | MMVet | POPE | DocV. | Chart. | Math. | Science. | AI2D |
| *Base Model* | | | | | | | | | | |
| LLaVA-Qwen2-7B | - | 74.44 | 46.67 | 38.85 | 84.55 | 50.62 | 52.72 | 38.00 | 74.91 | 73.77 |
| *Self-Improving Cognition* | | | | | | | | | | |
| SICOG-LLaVA-Qwen2-7B | SFT (Per., Rea.) | 75.45 | 48.60 | 37.84 | 84.35 | 52.52 | 54.48 | 38.80 | 77.44 | 76.20 |
| (Per., Rea., Lan.) | DPO (Per.), SFT (Rea.) | **76.18** | 48.40 | 38.72 | 83.53 | 52.20 | 54.80 | 39.20 | **77.49** | 75.78 |
| | DPO (Per., Rea.) | 74.83 | **49.00** | **38.90** | **84.85** | **52.54** | **55.64** | **41.00** | 76.20 | **76.33** |

**Preference learning is more effective than supervised fine-tuning for systematic perception and reasoning development.** Preference learning with DPO consistently surpasses standard supervised fine-tuning across all benchmarks for initializing systematic perception and reasoning in SICOG. For example, on MathVista, preference learning improves accuracy by approximately 2% on the low-resolution model LLaVA-Qwen2-7B, which is particularly challenging to enhance due to inherent

visual perception limitations. These results underscore the importance of learning not only from correct examples but also from avoiding mistakes, thereby fostering more robust skill development.

## F   HOW DOES SICOG ENHANCE THE REASONING CAPABILITIES OF FOUNDATION MLLMS?

Table 7: Evaluation results of SICOG variants on LLaVA-Qwen2-7B-UHD in two inference settings: direct answer and CoT for reasoning abilities.

| Method | Infer. | Train Data | Comprehensive | | | Hallu. | Chart/Table | | Knowledge | | |
|---|---|---|---|---|---|---|---|---|---|---|---|
| | | Stage 2 | MMBen. | MMStar | MMVet | POPE | DocV. | Chart. | Math. | Science. | AI2D |
| *Base Model* | | | | | | | | | | | |
| LLaVA-Qwen2-7B-UHD | Direct | - | 77.63 | 48.93 | 38.26 | 87.31 | 70.18 | 69.96 | 38.90 | 77.29 | 74.94 |
| *Self-Improving Cognition* | | | | | | | | | | | |
| | Direct | - | **77.80** | **52.47** | 40.14 | 87.84 | 73.05 | 72.24 | 41.40 | 79.42 | **78.40** |
| SICOG-LLaVA-Qwen2-7B-UHD (Perception, Reasoning, Language) | Direct | + Self-generated 45k VQA w/ DA&CoT | 76.12 | 51.00 | 39.82 | **88.02** | **74.15** | 73.12 | **42.90** | **80.61** | 78.21 |
| | CoT | + Self-generated 45k VQA w/ DA&CoT | 65.19 | 44.60 | **40.92** | 87.36 | 72.48 | **76.20** | 36.90 | 72.93 | 73.19 |

**Quantitative Analysis.**   We validate the efficacy of SICOG (perception, reasoning, language) in enhancing the reasoning capabilities of MLLMs under two inference settings: direct answer and CoT. The absence of CoT reasoning annotations in the instruction-tuning data (Zhang et al., 2024b; Liu et al., 2024a) used in stage 2 limits the model's ability to generate CoT reasoning. To address this limitation, we incorporate 45k self-generated visual instruction-tuning examples—originally used during the pre-training stage (stage 1.5)—into the instruction-tuning stage (stage 2) (see details in step 4 of section 3).

**Incorporating self-generated visual instruction-tuning data for instruction-tuning further improves multimodal reasoning.** As shown in Table 7, incorporating self-generated visual instruction-tuning data in stage 2 enhances SICOG's performance on most reasoning-intensive tasks across both inference settings. For instance, it provides an additional accuracy gain of approximately 1-4% on ChartQA. On benchmarks such as POPE and ScienceQA, direct answer inference outperforms CoT inference, likely due to the overwhelming prevalence of direct answer annotations compared to CoT annotations in the instruction-tuning data. In addition, we observe a performance drop on perception-heavy benchmarks like MMBench and MMStar. We suspect this decline stems from a data distribution shift introduced by the additional reasoning-focused data.

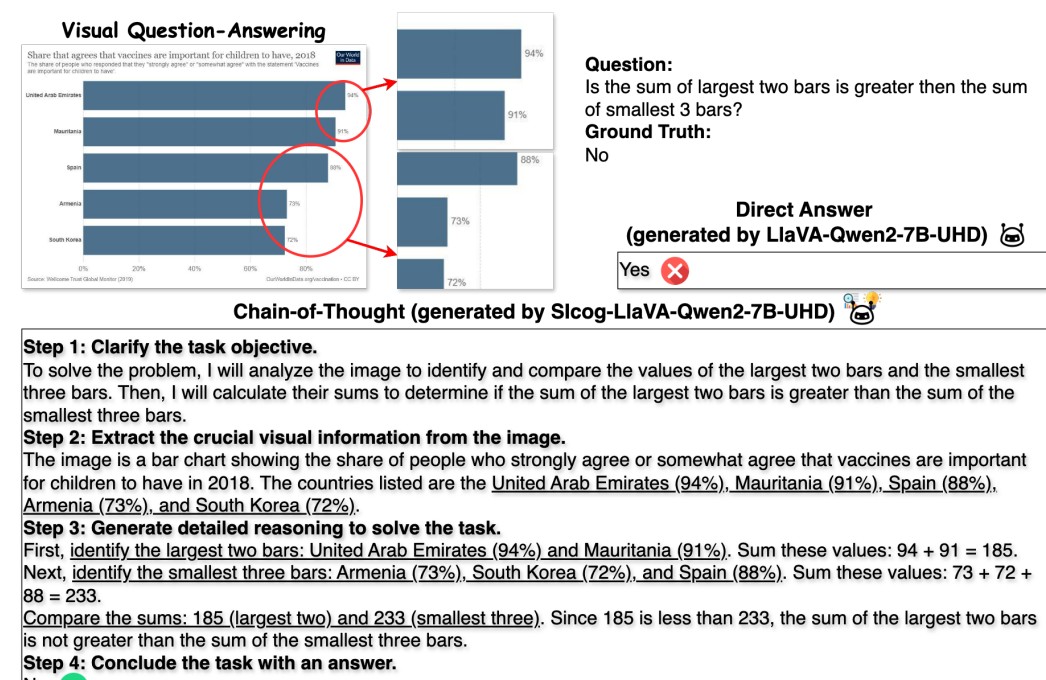

Figure 6: Qualitative comparison of responses generated by LLaVA-Qwen2-7B-UHD and SICOG-LLaVA-Qwen2-7B-UHD for the visual instruction-following task.

**Qualitative Analysis.** We compare the responses generated by the base LLaVA-Qwen2-7B-UHD and SICOG-LLaVA-Qwen2-7B-UHD (enhanced with self-generated VQA data in Stage 2) on an image-question pair from ChartVQA.

Figure 6 illustrates that, unlike LLaVA-Qwen2-7B-UHD, which produces an incorrect answer, SICOG-LLaVA-Qwen2-7B-UHD effectively integrates multimodal information into a systematic reasoning process, yielding an accurate and coherent response. Specifically, SICOG-LLaVA-Qwen2-7B-UHD first clarifies the task requirements and extracts key visual information, such as "United Arab Emirates (94%)" and "Mauritania (91%)". It then systematically leverages this information, identifying the two largest bars—United Arab Emirates (94%) and Mauritania (91%)—and the three smallest bars—Armenia (73%), South Korea (72%), and Spain (88%). By performing precise reasoning and calculations, SICOG-LLaVA-Qwen2-7B-UHD ultimately derives the correct answer. These findings confirm the efficacy of SICOG in enhancing the systematic reasoning capabilities of MLLMs, aligning with the results observed in the quantitative analysis.

## G    How Does Scaling Self-Generated Pre-Training Data Affect the Performance of SICOG?

The primary objective of multimodal pre-training is to refine and enhance knowledge acquisition from image captioning datasets (Li et al., 2024a). In this context, we analyze the effect of scaling self-generated pre-training data on SICOG-LLaVA-Qwen2-7B-UHD (perception, reasoning, language) by prioritizing an increase in the number of self-generated caption data. Specifically, we assess the performance of SICOG across four dimensions on ten benchmarks: comprehensive understanding (MM-Bench, MMStar, MMVet), hallucination (POPE), OCR and chart/table understanding (OCRBench, DocVQA, ChartQA), and knowledge-intensive tasks (MathVista, ScienceQA, AI2D) (Figure 7).

**Scaling up self-generated captions improves the performance of SICOG.** Increasing the quantity of self-generated caption data results in consistent performance improvements across three dimensions: comprehensive understanding (up to approximately 2%), OCR and chart/table understanding (up to around 2.5%), and knowledge-intensive tasks (up to around 3%), while maintaining stable performance on hallucination tasks. These improvements underscore the importance of scaling caption data in enhancing SICOG's ability to improve MLLMs' multimodal cognition. However, a slight performance decline occurs when the amount of caption data is increased without proportionally adjusting the quantities of visual and text-only instruction tuning data. We hypothesize that this decline arises from the overwhelming dominance of caption data, which creates an imbalanced data ratio and hinders effective model optimization.

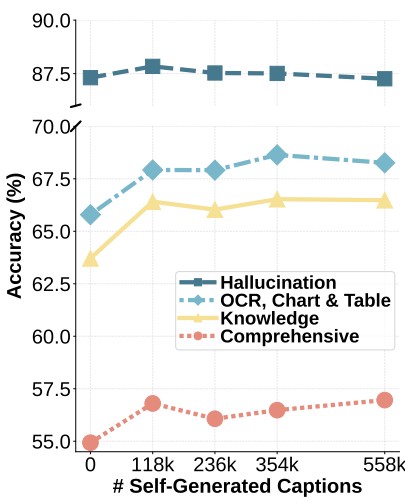

Figure 7: Impact of scaling self-generated captions on SICOG-LLaVA-Qwen2-7B-UHD during multimodal pre-training, evaluated across four dimensions on ten benchmarks.

## H    Does SICOG Remain Effective When Varying Recaptioned Images?

Table 8: Evaluation results of varying unlabeled image datasets for recaptioning on SICOG-LLaVA-Qwen2-7B-UHD across eleven benchmarks.

| Method | Comprehensive | | | Hallu. | Chart & Table | | | Knowledge | | | Vision |
|---|---|---|---|---|---|---|---|---|---|---|---|
| | MMBen. | MMStar | MMVet | POPE | OCR. | DocV. | Chart. | Math. | Science. | AI2D | Realworld. |
| *Base Model* | | | | | | | | | | | |
| LLaVA-Qwen2-7B-UHD | 77.63 | 48.93 | 38.26 | 87.31 | 55.20 | 70.18 | 69.96 | 38.90 | 77.29 | 74.94 | 63.53 |
| *Self-Improving Cognition* | | | | | | | | | | | |
| SICOG-LLaVA-Qwen2-7B-UHD (P., R., L.) (Recap. w/ BLIP 118k (Li et al., 2022)) | **77.80** | **52.47** | **40.14** | **87.84** | **57.70** | 73.05 | 72.24 | **41.40** | **79.42** | **78.40** | 63.92 |
| SICOG-LLaVA-Qwen2-7B-UHD (P., R., L.) (Recap. w/ V-FLAN 148k (Xu et al., 2024b)) | 77.07 | 51.93 | 38.67 | 87.50 | 56.40 | **73.45** | **73.60** | 40.20 | 79.38 | 77.85 | **67.19** |

We validate the effectiveness of SICOG across varying corpora by employing different unlabeled image datasets for recaptioning. Specifically, we randomly sample 148k images from the Vision-Flan (V-Flan) dataset (Xu et al., 2024b), which provides a diverse range of images and ensures zero overlap with the curated data described in Section 3.

**SICOG is robust to variations in recaptioned images.** As shown in Table 8, SICOG-LLaVA-Qwen2-7B-UHD (Perception, Reasoning, Language) consistently outperforms the base model, LLaVA-Qwen2-7B-UHD, achieving an approximate 4% accuracy gain on RealworldQA. This result

underscores the role of image diversity in enhancing real-world understanding and demonstrates the robust generalizability of SICOG across different corpora.

# I  CAN SICOG CONTRIBUTE TO THE CONSTRUCTION OF NEXT-GENERATION FOUNDATION MLLMS THROUGH CONTINUOUS COGNITIVE SELF-IMPROVEMENT?

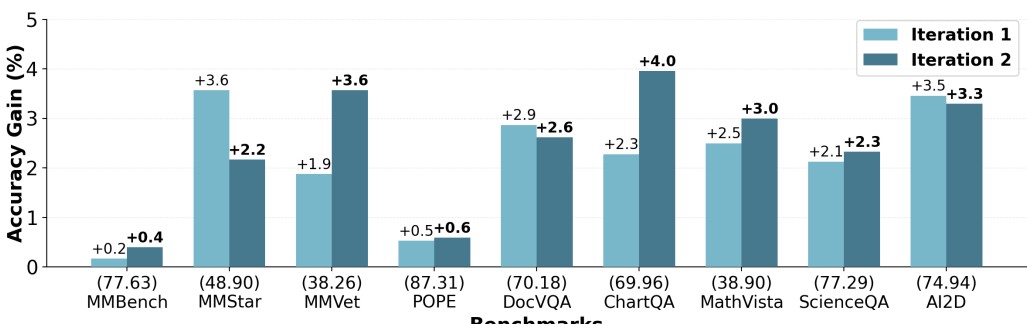

Figure 8: Evaluation results for next-generation foundation MLLM construction through continuous self-improvement using SICOG. Accuracy gains are reported as absolute improvements over the base model LLaVA-Qwen2-7B-UHD, with the base model's performance shown in parentheses.

The multimodal pre-training stage is specifically designed to expand the model's knowledge base Liu et al. (2024b). In this paper, we explore how self-learning can help expand the model's knowledge base during multimodal pre-training. We investigate the potential of SICOG in advancing next-generation foundational MLLM construction through continuous cognitive self-improvement. Specifically, we consider SICOG-LLaVA-Qwen2-7B-UHD (perception, cognition, language) from Table 1 as the foundational MLLM obtained in the first iteration. In the second iteration, 148K images from the V-Flan dataset are newly recaptioned, resulting in a self-curated dataset comprising caption data (118K from BLIP and 148K from V-Flan), VQA data (45K), and textual QA data (50K).

**SICOG drives next-generation foundational MLLM construction via continuous cognitive self-improvement.** Figure 8 presents absolute accuracy gains over the initial base MLLM, LLaVA-Qwen2-7B-UHD. The results show that SICOG-LLaVA-Qwen2-7B-UHD improves MLLM cognition across most benchmarks in the second iteration, achieving an additional 1.5% accuracy gain on MMVet. However, performance regressions on certain benchmarks may stem from an overrepresentation of caption data.

## J    HOW DO *Chain-of-Description* AND CHAIN-OF-THOUGHT IMPROVE COGNITION?

We examine two key factors underlying SICOG's efficacy: (*i*) the role of *Chain-of-Description* in facilitating multimodal perception, and (*ii*) the contribution of structured chain-of-thought to multimodal reasoning.

### J.1    HOW DOES *Chain-of-Description* FACILITATE MULTIMODAL PERCEPTION?

**Quantitative Analysis.**    We analyze captions for 100 images randomly sampled from BLIP-558k (Li et al., 2022), which is used as unlabeled image captioning data in Section 4. These captions are generated by perception-enhanced models fine-tuned on annotated caption data in three formats: detailed description (Detailed D), *Chain-of-Description* (CoD), and their combination (as implemented in SICOG, described in Section 3). Using GPT-4 with the prompt shown in Table 21, we evaluate six key dimensions: salient content, fine-grained details, relational attributes, peripheral content, faithfulness, and world knowledge. For a holistic analysis, we also include LLaVA-NeXT-34B, a leading open-source MLLM known for its strong captioning capabilities (Li et al., 2024a). Table 9 shows that the base model, regardless of resolution, consistently underperforms in salient content, fine-grained details, relational attributes, and peripheral content. These results highlight the importance of the four-step perception analysis design used in *Chain-of-Description*.

Table 9: Evaluation of re-captioning quality comparing the perception-enhanced models fine-tuned on curated caption data in three formats: detailed description (Detailed D), *Chain-of-Description* (CoD), and their combination (Section 3). Metrics (rated 1-5): salient content, fine-grained details, relational attributes, peripheral content, faithfulness, and world knowledge. "Caption": standard format; "Multi.": CoD step-by-step format (see Table 11 for details).

| Method | # Avg. Tokens | Systematic Perception | | | | General Performance | |
|---|---|---|---|---|---|---|---|
| | | Sali. | Fine-Grain. | Rela. | Peri. | Faith. | Know. |
| **Low-Resolution** | | | | | | | |
| LLaVA-Qwen2-7B | 129.36 | 4.51 | 4.21 | 3.82 | 3.67 | 4.07 | 3.63 |
| + Finetune w/ Detailed D | 129.68 | 4.59 | 4.49 | 3.88 | 3.88 | 4.13 | 3.87 |
| + Finetune w/ CoD (Caption) | 130.55 | 4.73 | 4.52 | 4.06 | 3.92 | 4.36 | 3.90 |
| + Finetune w/ CoD (Multi.) | **458.09** | 4.71 | 4.69 | 4.62 | 4.22 | 4.32 | 4.01 |
| + Finetune w/ Detailed D & CoD (Detailed D) | 130.13 | 4.75 | 4.54 | 4.13 | 3.97 | 4.49 | 3.95 |
| + Finetune w/ Detailed D & CoD (CoD Multi.) | 436.53 | **4.89** | **4.81** | **4.76** | **4.26** | **4.67** | **4.05** |
| **High-Resolution** | | | | | | | |
| LLaVA-Qwen2-7B-UHD | 135.08 | 4.77 | 4.30 | 3.99 | 3.81 | 4.41 | 3.84 |
| + Finetune w/ Detailed D | 140.73 | 4.71 | 4.52 | 3.92 | 3.91 | 4.20 | 3.77 |
| + Finetune w/ CoD (Caption) | 126.93 | 4.78 | 4.58 | 4.11 | 3.90 | 4.57 | 3.93 |
| + Finetune w/ CoD (Multi.) | 453.13 | 4.82 | 4.80 | 4.74 | 4.29 | 4.57 | 4.01 |
| + Finetune w/ Detailed D & CoD (Detailed D) | 136.50 | 4.76 | 4.67 | 4.01 | 3.82 | 4.51 | 3.88 |
| + Finetune w/ Detailed D & CoD (CoD Multi.) | **453.26** | **4.91** | **4.87** | **4.78** | **4.32** | **4.71** | **4.05** |
| LLaVA-NeXT-34B (Liu et al., 2024b) | 206.50 | 4.77 | 4.51 | 4.04 | 3.95 | 4.59 | 4.12 |

***Chain-of-Description* shows strong efficacy in facilitating systematic perception across six key dimensions.** Perception-enhanced models fine-tuned with *Chain-of-Description* outperform those trained on detailed descriptions in both single-step (caption-only) and multi-step formats. Notably, their combination achieves the highest evaluation scores, surpassing LLaVA-NeXT-34B in five of the six dimensions.

Furthermore, *Chain-of-Description* generates the longest average caption lengths (approximately 430–450 tokens), indicating a robust perceptual capacity. Additional analysis is provided in Appendix L.

**Qualitative Analysis.**    We compare two caption examples generated by LLaVA-Qwen2-7B-UHD and perception-enhanced LLaVA-Qwen2-7B-UHD (adopted in SICOG) on an image from V-FLAN 148k, as referenced in Table 8.

In Figure 9, the results reveal that *Chain-of-Description* enables MLLMs to capture richer and more detailed visual information across all six dimensions, whereas captions generated by the base LLaVA-Qwen2-7B-UHD often include inaccuracies, *e.g.,* hallucinations (Bai et al., 2024). For instance, *Chain-of-Description* allows MLLMs to identify nuanced details such as "the road appears wet" and "the lighting conditions are subdued" in step 4, suggesting "recent rain" and an "overcast day." In contrast, the base LLaVA-Qwen2-7B-UHD fails to capture these details, resulting in an inaccurate description of "a clear sky." This observation aligns with quantitative findings, confirming *Chain-of-Description*'s effectiveness in enhancing systematic perception.

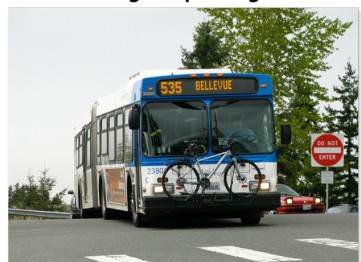

**Image Captioning**

**Detailed Description (generated by LlaVA-Qwen2-7B-UHD)** 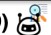

The image captures a blue city bus on a roadway, prominently displaying the route number "535" and destination "BELLEVUE" at the top front. The bus features large windows, a windshield with wipers, and headlights, suggesting it is designed for public transit. Mounted on the front are two bicycles, indicating the bus accommodates bike-friendly policies. A visible license plate reads "238024". The bus is in motion, as indicated by its forward position relative to the road markings. In the background, a red car is slightly blurred, suggesting motion or a slower shutter speed. A sign reading "DO NOT ENTER" is partially visible on the right side, indicating traffic restrictions. The setting appears to be a suburban area with trees and a clear sky. ❌

*Chain-of-Description* (generated by perception-enhanced LlaVA-Qwen2-7B-UHD)

**Step 1: Extract salient content by identifying the key elements that define the image.**
The image shows a large public transportation bus positioned on a roadway. The bus is predominantly blue with white accents and displays the route number "535" along with the destination "BELLEVUE" at the top front. A bicycle is mounted on the front bike rack of the bus, which is currently empty.
**Step 2: Analyze detailed information, focusing on instance-level attributes such as low-level and fine-grained details.**
The bus's design includes a large windshield and multiple windows along its side. The bike rack is clearly visible, holding the bicycle securely. The bus is in motion, as indicated by the blurred background, suggesting the photo was taken while the bus was traveling.
**Step 3: Consider relational-level attributes, analyzing interactions between elements and their spatial organization.**
In the background, there is a red car partially visible behind the bus, and a road sign reading "DO NOT ENTER" is prominently displayed to the right. The scene is set against a backdrop of trees and an overcast sky, contributing to the overall atmosphere of the image.
**Step 4: Examine marginal or peripheral content to ensure no important information is missed.**
The road appears wet, suggesting recent rain, and the lighting conditions are subdued, indicating an overcast day. These environmental details enhance the context of the image, providing insight into the setting of day.
**Step 5: Organize all observations into a detailed, cohesive description.**
The image depicts a large public transportation bus on a roadway, predominantly painted blue with white accents. The bus has the route number "535" and destination "BELLEVUE" displayed at the top front. A bicycle is mounted on the front bike rack of the bus, which is currently empty. The bus features a large windshield and multiple side windows. In the background, a red car is partially visible behind the bus, and a road sign reading "DO NOT ENTER" is prominently displayed to the right. The scene is set against a backdrop of trees and an overcast sky, with the road appearing wet, suggesting recent rain. The lighting conditions indicate an overcast day. ✅

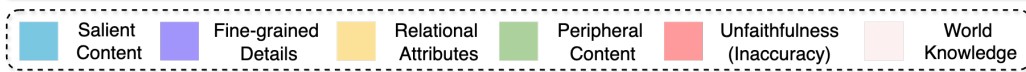

| Salient Content | Fine-grained Details | Relational Attributes | Peripheral Content | Unfaithfulness (Inaccuracy) | World Knowledge |

Figure 9: Qualitative comparison of captions generated by LLaVA-Qwen2-7B-UHD and perception-enhanced LLaVA-Qwen2-7B-UHD across six key dimensions in the image captioning task.

*Chain-of-Description* **avoids redundancy through systematic step-wise analysis.** While *Chain-of-Description* generates comprehensive and detailed descriptions, it maintains efficiency by *eliminating information overlap* in its step-by-step analysis. For instance, peripheral elements such as "a red car" and "a road sign reading 'do not enter'" are described with precise spatial relations in step 3. Subsequently, step 4 focuses exclusively on new observations, such as "the road appears wet," ensuring non-redundant content progression and avoiding verbosity.

## J.2 HOW DOES STRUCTURED CHAIN-OF-THOUGHT ENHANCE MULTIMODAL REASONING?

**Quantitative Analysis.** We evaluate the accuracy of answers for 1k image-question pairs randomly sampled from the 63k `LLaVA-CoT` split, which is used as unlabeled pre-training VQA data in Section 4. These answers are generated by reasoning-enhanced models fine-tuned on curated

reasoning data in three formats: direct answer, structured CoT, and their combination (as implemented in SICOG). As shown in Table 10, models fine-tuned with structured CoT improve the base models' performance by 9% in EM. The combination of CoT and direct answer achieves the best results, outperforming other methods by 2% to 5% EM, demonstrating the effectiveness of structured CoT in enhancing multimodal reasoning in MLLMs. However, models fine-tuned solely with CoT perform comparably or slightly worse than those fine-tuned with direct answers, likely due to insufficient CoT reasoning data in the base models' training set.

Table 10: Evaluation of self-generated reasoning quality, comparing reasoning-improved models fine-tuned on curated reasoning data in three formats: direct answer (Direct), chain-of-thought (CoT), and their combination (Section 3). Exact Match (EM) scores are used to assess the correctness of final answers.

| Low-Resolution | | High-Resolution | |
|---|---|---|---|
| **Method** | **Correct. (EM)** | **Method** | **Correct. (EM)** |
| LLaVA-Qwen2-7B | 26 | LLaVA-Qwen2-7B-UHD | 33 |
| + Finetune w/ Direct Answer | 35 | + Finetune w/ Direct Answer | 43 |
| + Finetune w/ Chain-of-Thought (CoT) | 35 | + Finetune w/ Chain-of-Thought (CoT) | 42 |
| + Finetune w/ Direct Ans. & CoT (Direct) | **37** | + Finetune w/ Direct Ans. & CoT (Direct) | **47** |
| + Finetune w/ Direct Ans. & CoT (CoT) | **37** | + Finetune w/ Direct Ans. & CoT (CoT) | **47** |

**Qualitative Analysis.** We compare the responses generated by the base LLaVA-Qwen2-7B-UHD and the reasoning-enhanced LLaVA-Qwen2-7B-UHD (used in SICOG) on an image-question pair from the 63k `LLaVA-CoT` split.

In Figure 10, our analysis demonstrates that the structured CoT enables the MLLM to generate systematic, logical, and in-depth reasoning step-by-step, resulting in accurate answers. Specifically, CoT first helps clarify the task requirements and captures critical visual information. Then, CoT enables the MLLM to utilize key visual details, such as "212.22 million U.S. dollars" and "354 million U.S. dollars," to perform accurate reasoning and calculations.

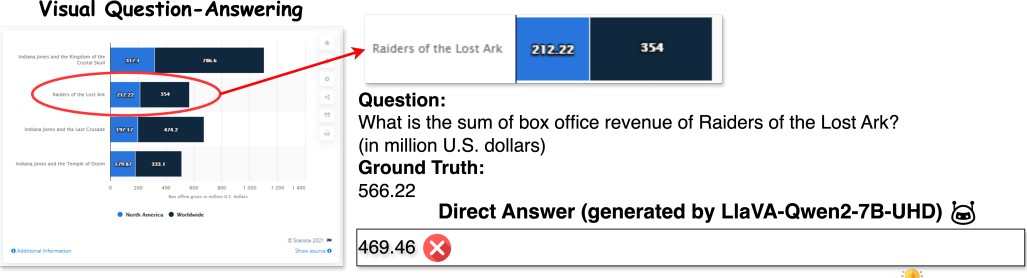

Figure 10: Qualitative comparison of responses generated by LLaVA-Qwen2-7B-UHD and reasoning-enhanced LLaVA-Qwen2-7B-UHD in the visual question-answering task.

## K   FINE-GRAINED EVALUATION ACROSS SIX CORE CAPABILITIES

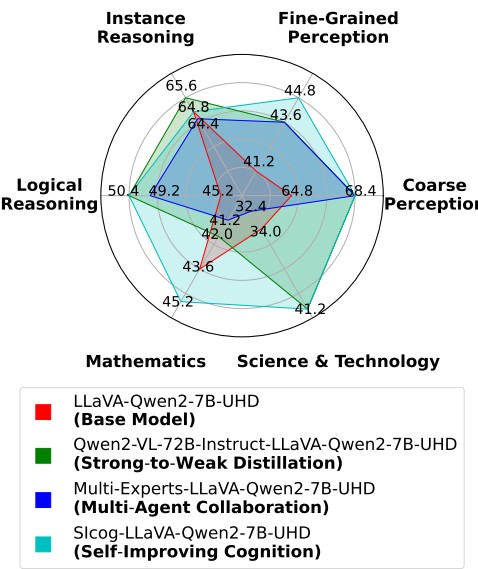

Figure 11: Fine-grained evaluation of six core capabilities on LLaVA-Qwen2-7B-UHD using the MMStar benchmark (direct answer).

The fine-grained evaluation of six core capabilities in Figure 11 highlights the effectiveness of SICOG in advancing multimodal cognitive abilities in MLLMs.

## L   DISTRIBUTION OF GENERATED CAPTION TOKEN LENGTHS

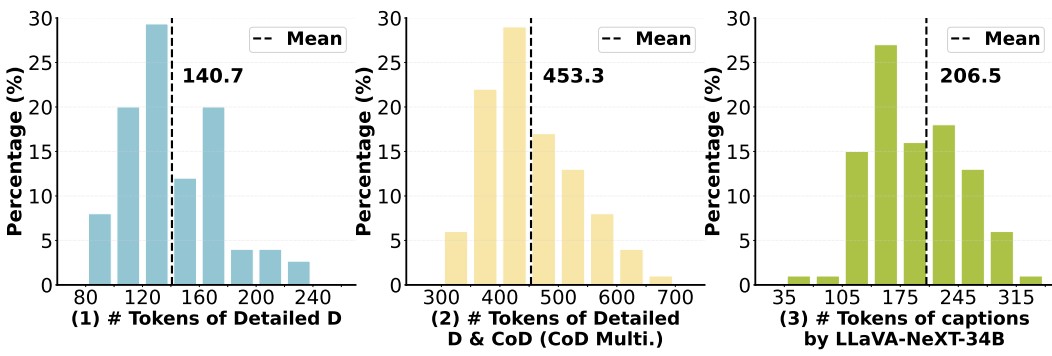

Figure 12: Distribution of caption token lengths: perception-enhanced LLaVA-Qwen2-7B-UHD (fine-tuned on curated captions using the standard detailed description format and the mixture of detailed description and *Chain-of-Description* formats) *vs.* LLaVA-NeXT-34B.

Figure 12 compares token length distributions, showing that *Chain-of-Description* consistently produces longer captions across all images, suggesting a robust perceptual ability.

## M  IMPLEMENTATION DETAILS

Table 11: Three different caption formats used for Chain-of-Description conversion during the multimodal pre-training stage (1/2).

---

**Three Caption Formats for Chain-of-Description Conversion in Multimodal Pre-Training**

---

**Format 1: Standard caption-only.**

---

**Human:** <image>\nPlease provide a thorough description of this image.
**Model:** The image showcases a modern spiral staircase with a unique design. The staircase features a curved, wooden handrail that is dark brown in color, contrasting with the lighter wooden steps. The railings are made of the same dark wood and have a horizontal slat design, giving a sense of movement along the staircase's curve. The staircase itself appears to be constructed from polished wood, giving it a sleek and contemporary look. Below the staircase, the flooring is light beige, complementing the warm tones of the wooden staircase. The room surrounding the staircase is spacious, with large windows that allow natural light to enter, enhancing the brightness of the space. There are no people or objects in the immediate vicinity of the staircase, emphasizing the architectural features. The overall color scheme is warm, with the wood tones being the dominant feature.

---

**Format 2: Step-by-step elicitation.**

---

**Human:** <image>\nPlease provide a thorough description of this image. Explain your description step by step.
**Model:** Step 1: Extract salient content by identifying the key elements that define the image.
The image depicts a modern spiral staircase with a striking design. The staircase features a curved, wooden handrail that is dark brown in color, contrasting with the lighter wooden steps. The railings are made of the same dark wood and have a horizontal slat design, providing a sense of movement along the staircase's curve.
Step 2: Analyze detailed information, focusing on instance-level attributes such as low-level and fine-grained details.
The staircase is constructed from polished wood, which gives it a sleek and contemporary appearance. The flooring below the staircase is light beige, complementing the warm tones of the wooden staircase. The room surrounding the staircase is spacious, with large windows that allow natural light to enter, enhancing the brightness of the space.
Step 3: Consider relational-level attributes, analyzing interactions between elements and their spatial organization.
The staircase's design is both functional and aesthetic, with its spiral form creating a dynamic visual element within the room. The natural light from the windows highlights the warm tones of the wood and adds depth to the space. The absence of people or objects in the room emphasizes the architectural features of the staircase.
Step 4: Examine marginal or peripheral content to ensure no important information is missed.
The room's architecture is modern, with clean lines and minimalistic design elements. The combination of the staircase's curved form and the natural light creates an inviting and open atmosphere in the space.
Step 5: Organize all observations into a detailed, cohesive description.
The image showcases a modern spiral staircase with a unique design. The staircase features a curved, wooden handrail that is dark brown in color, contrasting with the lighter wooden steps. The railings are made of the same dark wood and have a horizontal slat design, giving a sense of movement along the staircase's curve. The staircase itself appears to be constructed from polished wood, giving it a sleek and contemporary look...

---

### M.1  COMPARED METHODS.

We compare SICOG against the following representative MLLM pre-training approaches (as discussed in Section 2). Differences are considered significant at $p < 0.01$:

- **Strong-to-Weak Distillation (Perception)** (Li et al., 2024a): Pre-training with re-caption data containing detailed descriptions (DD) generated by stronger models.

- **Multi-Agent Collaboration (Perception)** (Fang et al., 2024): Pre-training with re-caption data containing detailed descriptions and fine-grained attributes (DD-FGA) generated by base and expert models.

Table 12: Three different caption formats used for Chain-of-Description conversion during the multimodal pre-training stage (2/2).

| Three Caption Formats for Chain-of-Description Conversion in Multimodal Pre-Training |
| --- |
| **Format 3: Multi-turn conversation.** |
| **Human:** <image>\nWhat are the crucial details that define the image? 
 **Model:** The image depicts a modern spiral staircase with a striking design. The staircase features a curved, wooden handrail that is dark brown in color, contrasting with the lighter wooden steps. The railings are made of the same dark wood and have a horizontal slat design, providing a sense of movement along the staircase's curve. 
 **Human:** Can you analyze the image for instance-level attributes and low-level details? 
 **Model:** The staircase is constructed from polished wood, which gives it a sleek and contemporary appearance. The flooring below the staircase is light beige, complementing the warm tones of the wooden staircase. The room surrounding the staircase is spacious, with large windows that allow natural light to enter, enhancing the brightness of the space. 
 **Human:** What is the relationship between the components, and how are they arranged? 
 **Model:** The staircase's design is both functional and aesthetic, with its spiral form creating a dynamic visual element within the room. The natural light from the windows highlights the warm tones of the wood and adds depth to the space. The absence of people or objects in the room emphasizes the architectural features of the staircase. 
 **Human:** Is there anything in the margins or borders of the image worth noting? 
 **Model:** The room's architecture is modern, with clean lines and minimalistic design elements. The combination of the staircase's curved form and the natural light creates an inviting and open atmosphere in the space. 
 **Human:** How would you describe the image in a well-organized and cohesive manner? 
 **Model:** The image showcases a modern spiral staircase with a unique design. The staircase features a curved, wooden handrail that is dark brown in color, contrasting with the lighter wooden steps. The railings are made of the same dark wood and have a horizontal slat design, giving a sense of movement along the staircase's curve. The staircase itself appears to be constructed from polished wood, giving it a sleek and contemporary look... |

Table 13: Hyperparameters, training configurations, and inference time for the SICOG implementation in Step 4 (Section 3). Inference time is reported by data generation mode: Detailed Descriptions (D) and Direct Answers (A) on unlabeled images from the 118k BLIP dataset, and Chain of Thought (CoT) and CoD on unlabeled image–question pairs from the 63k LLaVA-CoT dataset.

| Model | LLaVA-Qwen2-7B | | | LLaVA-Qwen2-7B-UHD | | | LLaVA-Llama3.1-8B-UHD | | |
| --- | --- | --- | --- | --- | --- | --- | --- | --- | --- |
| **Training Stage** | Stage 1 | Stage 1.5 | Stage 2 | Stage 1 | Stage 1.5 | Stage 2 | Stage 1 | Stage 1.5 | Stage 2 |
| Learning Rate | 2e-4 | 2e-5 | 2e-5 | 2e-4 | 2e-5 | 2e-5 | 2e-4 | 2e-5 | 2e-5 |
| Batch Size | 256 | 128 | 128 | 256 | 128 | 128 | 256 | 128 | 128 |
| Sequence Length | 4096 | 4096 | 4096 | 4096 | 4096 | 4096 | 4096 | 4096 | 4096 |
| Epochs | 1 | 1 | 1 | 1 | 1 | 1 | 1 | 1 | 1 |
| Training Time | 50 min | 40 min | 5.2 h | 1 h | 1.5 h | 9.5 h | 1 h | 1.8 h | 12.3 h |
| Resource (GPUs) | 4×8 NVIDIA A100 80GB | | | 4×8 NVIDIA A100 80GB | | | 4×8 NVIDIA A100 80GB | | |
| **Inference Time (Self-Generating Pre-training Data)** | | | | | | | | | |
| Detailed Descriptions / CoD | 4 h / 12.5 h | | | 4 h / 13 h | | | 3 h / 6.5 h | | |
| Direct Answers / CoT | 2 h / 8.5 h | | | 1.5 h / 9 h | | | 1.5 h / 9.5 h | | |
| Resource (GPUs) | 4×8 NVIDIA A100 80GB | | | 4×8 NVIDIA A100 80GB | | | 4×8 NVIDIA A100 80GB | | |

- **Self-Improving Cognition (Perception & Reasoning – Ours):** Pre-training with self-generated data, including re-caption data containing detailed descriptions (DD) and *Chain-of-Description* (CoD), visual instruction-tuning data with direct answers (DA) and structured CoT, as well as text-only instruction-tuning data.

## M.2 IMPLEMENTATION DETAILS.

(*i*) *Models:* We utilize both low-resolution (LLaVA-Qwen2-7B (Liu et al., 2023)) and high-resolution (LLaVA-Qwen2-7B-UHD (Guo et al., 2024) and LLaVA-Llama3.1-8B-UHD) models for our investigation. Specifically, we employ CLIP-ViT-L/14-336 (Radford et al., 2021) as the visual encoder,

Qwen2-7B-Instruct (Yang et al., 2024a) and LLama-3.1-8B-Instruct (Grattafiori et al., 2024) as the backbone LLMs.

(*ii*) *Self-Generated Data Source:* We recaption the images used for modality alignment and self-annotate 63k image-question pairs randomly selected from `LLaVA-CoT`, ensuring zero overlap with the curated training data in Section 3. Additionally, we self-annotate 100k text-only prompts randomly sampled from OpenHermes-2.5 (Teknium, 2023).

(*iii*) *Data Utilized in the Three-Stage Training Strategy:* During the self-refinement step (Step 4 in Figure 2), we use BLIP-558k caption data (Radford et al., 2021) for modality alignment, following (Liu et al., 2024b). For multimodal pre-training, we use self-generated data along with 858k instruction-tuning samples organized by (Zhang et al., 2024b), which include the widely adopted LLaVA-Mix665k (Liu et al., 2024a) and 160k samples from UReader (Ye et al., 2023).

(*iv*) *Implementing Step 2 of* SICOG*:* To collect pre-training data, we prompt the model multiple times to sample candidate outputs using a temperature of 0.7 and top-p of 0.95:

- For image captioning, we sample three candidate captions (two in *Chain-of-Description* format and one as a detailed description).
- For visual instruction tuning, we sample three candidate responses (two in chain-of-thought (CoT) format and one as a direct answer).
- For text-only instruction tuning, we sample three candidate responses.

(*v*) *Implementing Step 3 of* SICOG*:* To ensure the quality of self-generated pre-training data, we use NV-Embed-v2 (Lee et al., 2024) to generate candidate embeddings and calculate their semantic similarity. The data is curated based on similarity scores and predefined thresholds. Specifically, we apply the following curation strategies:

- **Curation of Self-Generated Image Captioning Data:** We set the similarity threshold $\tau^{\text{Perception}} = 0$ and retain all top-1 ranked self-generated captions in a mixture of two formats: detailed descriptions and *Chain-of-Description*. To preserve the MLLM's multi-turn conversational ability, we convert the top-1 ranked *Chain-of-Description* captions with a consistency score higher than $0.85$ into a multi-turn conversational format. Detailed examples are provided in Tables 11 and 12.

  In addition, Table 14 presents a comparison of two perception-enhancement approaches for SICOG: (*i*) fine-tuning with 35 detailed descriptions and (*ii*) *Chain-of-Description* in three formats (standard caption-only, step-by-step elicitation, and multi-turn conversation, as shown in Tables 11 and 12). The results suggest: (1) the efficacy of the proposed *Chain-of-Description* approach in enhancing richer visual understanding. (2) the impact of the three different *Chain-of-Description* formats on the overall performance of SICOG.

- **Curation of Self-Generated Visual Instruction-Tuning Data:** We set the similarity threshold $\tau^{\text{Perception}} = 0.95$ and retain all top-1 ranked self-generated responses in two formats: direct answers and chain-of-thought (CoT).

- **Curation of Self-Generated Text-Only Instruction-Tuning Data:** We set the similarity threshold $\tau^{\text{Perception}} = 0.8$ and retain only the first 50k text-only prompt-response pairs based on their similarity score rankings.

Extensive experiments and analyses are conducted using 128 A100 80G GPUs. Table 13 summarizes the hyperparameters used to implement SICOG in Step 4 (Section 3); the same settings are also applied in Step 1 to develop the MLLM's capabilities during Stage 2.

### M.3 IMPLEMENTATION DETAILS OF COMPARED METHODS.

(*i*) *Implementing Weak-to-Strong Distillation:* Following (Li et al., 2024a), we prompt Qwen2-VL-72B-Instruct (Wang et al., 2024) and LLaVA-NeXT-34B (Liu et al., 2024b), two leading open-source MLLMs known for their strong captioning capabilities, to generate high-quality captions for unlabeled images. These captions are used as multi-modal pre-training data to construct smaller foundation MLLMs, resulting in models such as Qwen2-VL-72B-Instruct-LLaVA-Qwen2-7B-UHD, LLaVA-NeXT-34B-LLaVA-Qwen2-7B-UHD, and others. Generating captions for 118k unlabeled BLIP

Table 14: Comparison of two perception-enhancement approaches for SICOG: (*i*) fine-tuning with 35k detailed descriptions and (*ii*) *Chain-of-Description* in three formats (as shown in Tables 11 and 12). Refer to Table 9 for a detailed discussion. The self-generated caption data used are sampled only once, without filtering.

| Method | Perception Enhancement | Train Data | Comprehensive | | Hallu. | Chart/Table | | Knowledge | | |
|---|---|---|---|---|---|---|---|---|---|---|
| | | Stage 1.5 | MMBen. | MMStar | POPE | DocV. | Chart. | Math. | Science. | AI2D |
| *Base Model* | | | | | | | | | | |
| LLaVA-Qwen2-7B-UHD | - | - | 77.63 | 48.93 | **87.31** | 70.18 | 69.96 | 38.90 | 77.29 | 74.94 |
| | Fintune w/ 35k DD | Self-generated 118k caption w/ DD | 77.19 | 50.67 | 85.74 | 71.43 | 71.96 | 37.70 | 78.14 | 75.94 |
| | | Self-generated 118k caption w/ CoD (caption) | 78.25 | **50.93** | 86.80 | 71.48 | 72.12 | 40.30 | 77.84 | **76.81** |
| SICOG-LLaVA-Qwen2-7B-UHD (Perception) | | Self-generated 118k caption w/ CoD (multi.) | 78.48 | 50.20 | 86.22 | 71.74 | **72.52** | 41.20 | 76.95 | 76.33 |
| | Fintune w/ 35k CoD | Self-generated 118k caption w/ CoD (conv.) | **78.53** | 50.87 | 87.19 | 71.67 | 72.16 | 40.00 | **78.24** | 76.62 |
| | | Self-generated 118k caption w/ CoD (caption, multi., conv.) | 77.52 | 50.60 | 86.86 | **71.84** | 72.32 | **41.60** | 77.44 | 76.75 |

images using Qwen2-VL-72B-Instruct requires approximately 110 hours on a cluster of 4×8 NVIDIA A100 80GB GPUs. To facilitate reproducibility, we directly utilize the open-sourced recaptioned dataset generated by LLaVA-NeXT-34B (Liu et al., 2024b).

(*ii*) *Implementing Multi-Agent Collaboration:* Following (Fang et al., 2024), we employ three specialized modules—Spatial Specialist, OCR Specialist, and Grounding Specialist—to extract fine-grained attributes. These attributes are combined with self-generated detailed descriptions to form rich pre-training captions. Specifically, we prompt Qwen2-VL-72B-Instruct to generate spatial attributes using the instruction: *"Elaborate on the visual and narrative elements of the image in detail, with a focus on spatial relations."* This annotation process, applied to the 118k unlabeled BLIP images, requires approximately 100 hours on a cluster of 4×8 NVIDIA A100 80GB GPUs. For OCR-based annotations, we utilize PaddleOCR (PaddlePaddle Team, 2020), retaining only outputs with a confidence score above 0.9. For object grounding, we adopt GroundingDINO (Liu et al., 2024c) with a detection threshold of 0.435. The OCR and grounding annotation processes take approximately 1 hour on 8 NVIDIA A100 80GB GPUs. After extracting all attributes, we use Qwen2-VL-72B-Instruct to rephrase the outputs for improved fluency and clarity. Finally, we concatenate the self-generated detailed descriptions with the spatial, OCR, and grounding attributes to construct multi-turn conversational caption data, following the methodology of (Fang et al., 2024).

# N ANALYSIS OF COMPUTATIONAL COST

Table 15: Computational analysis of the compared pre-training approaches for generating the pre-training data.

| Method | Generation Task | Model(s) Used | Hardware (A100 80G) | Time (Hours) | Est. GPU-Hours (Total) |
|---|---|---|---|---|---|
| ***External Model Distillation*** | | | | | |
| Strong-to-Weak Distillation | Detailed Descriptions | Qwen2-VL-72B-Instruct | 32 GPUs | ~110 | ~3,520 |
| Multi-Agent Collaboration | Spatial Attributes | Qwen2-VL-72B-Instruct | 32 GPUs | ~100 | ~3,200 |
| | OCR Annotation | PaddleOCR | 8 GPUs | ~1 | ~8 |
| | Grounding Information | GroundingDINO | 8 GPUs | ~1 | ~8 |
| | Detailed Descriptions | Qwen2-VL-72B-Instruct | 32 GPUs | ~115 | ~3,680 |
| ***Self-Improving (Ours)*** | | | | | |
| SICOG | Post-Training Optimization | LLaVA-Qwen2-7B-UHD | 32 GPUs | ~2 | ~64 |
| | Detailed Captions (DD & CoD) | LLaVA-Qwen2-7B-UHD | 32 GPUs | ~21 | ~672 |
| | VQA Responses (DA & CoT) | LLaVA-Qwen2-7B-UHD | 32 GPUs | ~12 | ~384 |
| | Text-Only Responses | Qwen2-7B-Instruct | 8 GPUs | ~8 | ~64 |
| | Candidate Filtering (Captions, VQA, Text-only) | NV-Embed-v2 | 8 GPUs | ~20 | ~160 |

The core difference between pre-training approaches lies in how multimodal pre-training data is obtained. Table 15 presents a computational cost analysis of the compared approaches for generating multimodal pre-training data. While SICOG involves more training stages compared to prevalent approaches that rely on external advanced or expert models, it significantly reduces computational cost in terms of GPU hours.

We would also like to emphasize that SICOG is not designed to directly compete with existing pre-training methods that depend on external annotations. Instead, it is tailored for scenarios where high-quality external annotations are unavailable—such as when model capabilities surpass human performance, when obtaining annotations is prohibitively expensive, or when no stronger model exists to provide annotations. Additionally, SICOG can complement methods that leverage external annotations when such resources are accessible.

# O   MITIGATING POTENTIAL PERFORMANCE SATURATION AND ERROR PROPAGATION

Table 16: Exploration of Mitigating Potential Performance Saturation

| Method | Comprehensive | | Hallu. | Chart/Table | | Knowledge | | |
| --- | --- | --- | --- | --- | --- | --- | --- | --- |
| | MMBen. | MMStar | POPE | DocV. | Chart. | Math. | Science. | AI2D |
| LLaVA-Qwen2-7B-UHD | 77.63 | 48.93 | 87.31 | 70.18 | 69.96 | 38.90 | 77.29 | 74.94 |
| Self-Caption-LLaVA-Qwen2-7B-UHD | 77.41 | 49.30 | 86.67 | 70.16 | 71.32 | 38.90 | 76.40 | 75.87 |
| SICOG-LLaVA-Qwen2-7B-UHD | **78.08** | **52.47** | **87.84** | **73.70** | **73.12** | **41.40** | **79.42** | **78.40** |

Table 17: Exploration of Mitigating Error Propagation

| Method | Comprehensive | | Hallu. | Chart/Table | | Knowledge | | |
| --- | --- | --- | --- | --- | --- | --- | --- | --- |
| | MMBen. | MMStar | POPE | DocV. | Chart. | Math. | Science. | AI2D |
| LLaVA-Qwen2-7B-UHD | 77.63 | 48.93 | 87.31 | 70.18 | 69.96 | 38.90 | 77.29 | 74.94 |
| SICOG-LLaVA-Qwen2-7B-UHD w/o Filtering | 77.97 | 50.87 | 87.56 | 72.97 | 73.64 | 39.50 | 77.84 | 76.98 |
| SICOG-LLaVA-Qwen2-7B-UHD | **78.08** | **52.47** | **87.84** | **73.70** | **73.12** | **41.40** | **79.42** | **78.40** |

To mitigate the potential risks of relying on the model to generate its own data, particularly in terms of performance saturation and error propagation. To mitigate these risks, we have implemented the following measures:

*Mitigating Performance Saturation.* Our framework enhances the model's perception and reasoning abilities using minimal annotations during Stage 1. By ensuring meaningful improvements in the base model before initiating self-improvement iterations, we establish a robust self-improving paradigm. Below, we provide results leveraging the model's self-generated detailed captions as pre-training data (since the base model cannot generate chain-of-thought reasoning). This variant is referred to as "Self-Caption-LLaVA-Qwen2-7B-UHD." In Table 16, we observe that SICOG-LLaVA-Qwen2-7B-UHD achieves superior performance, highlighting the value of fine-tuning with minimal annotated data during Stage 1 to ensure performance improvements.

*Avoiding Error Propagation.* We employ a robust data filtering mechanism to select high-quality self-generated data for pre-training, as detailed in Section 3. This minimizes the impact of noisy or biased outputs. Below, we present results after removing the filtering mechanism (denoted as "w/o Filtering"). In Table 17, SICOG-LLaVA-Qwen2-7B-UHD demonstrates improved performance when the filtering mechanism is applied, underscoring its importance in enhancing the self-improving paradigm.

While these measures address the immediate risks, we acknowledge the importance of further refinement. We aim for SICOG to serve as a starting point for advancing self-improvement techniques and are committed to exploring additional strategies to ensure robustness in future iterations.

# P  ADDITIONAL EXPERIMENTAL RESULTS

Table 18: Additional evaluation results on MMMLU-Val (Yue et al., 2024a).

| Method | Type | MMMLU |
|---|---|---|
| LLaVA-Qwen2-7B-UHD | - | 55.38 |
| Qwen2-VL-72B-Instruct-LLaVA-Qwen2-7B-UHD | Strong-to-Weak Distillation | 52.25 |
| Multi-Experts-LLaVA-Qwen2-7B-UHD | Multi-Experts | 52.25 |
| SICOG-LLaVA-Qwen2-7B-UHD | Self-Learning | **56.88** |

Table 19: Comparison with other pre-training methods.

| Method | Comprehensive | | Hallu. | Chart/Table | | Knowledge | | |
|---|---|---|---|---|---|---|---|---|
| | MMBen. | MMStar | POPE | DocV. | Chart. | Math. | Science | AI2D |
| LLaVA-Qwen2-7B-UHD | 77.63 | 48.93 | 87.31 | 70.18 | 69.96 | 38.90 | 77.29 | 74.94 |
| Self-Caption- | 77.41 | 49.30 | 86.67 | 70.16 | 71.32 | 38.90 | 76.40 | 75.87 |
| LLaVA-NeXT-34B-Caption- | 77.75 | 50.60 | 86.46 | 71.20 | 71.56 | 36.90 | 78.38 | 76.00 |
| LLaVA-NeXT-34B-Caption-GPT-4o-Reason- | 77.80 | 51.27 | 87.00 | 72.20 | 72.80 | 37.80 | 78.93 | 77.49 |
| SICOG-LLaVA-Qwen2-7B-UHD | **78.08** | **52.47** | **87.84** | **73.70** | **73.12** | **41.40** | **79.42** | **78.40** |

**Results on MMMLU.** Due to space limitations, we present the results on the MMMLU-Val dataset in Table 18.

**Comparison with other pre-training methods.** Existing open-source models lack the ability for systematic perception and reasoning. Specifically, directly prompting these models to generate Chain-of-Description (CoD) captions and structured Chain-of-Thought (CoT) reasoning data (Xu et al., 2025) is impractical. Furthermore, none of the backbone models used in this work possess multimodal CoT reasoning abilities, as their training data does not include multimodal CoT reasoning examples. This provides a clear framework to evaluate how incorporating self-generated CoT data during pre-training impacts model performance.

We also provide additional results in Table 19:

*1. Comparison with pre-training using re-captioned data generated via prompting the base model (Self-Caption-LLaVA-Qwen2-7B-UHD).*

*2. Comparison with pre-training using re-captioned data from a stronger captioning model and CoT reasoning data.* Specifically, we use LLaVA-NeXT-34B for captioning and GPT-4o for CoT reasoning data, curated in Xu et al. (2025) (LLaVA-NeXT-34B-Caption-GPT-4o-Reason-LLaVA-Qwen2-7B-UHD).

Our approach, SICOG-LLaVA-LLaVA-Qwen2-7B-UHD, outperforms these methods, demonstrating the effectiveness of incorporating self-generated CoT data during pre-training.

# Q  PROMPTS

Table 20: The prompt utilized by GPT-4o for eliciting *Chain-of-Description* for image-captioning training datasets.

---

**Prompt Used by GPT-4o for eliciting *Chain-of-Description***

You are an expert AI assistant tasked with analyzing an image and generating a detailed, step-by-step description. You are provided with an original description as a reference. Your goal is to ensure accuracy, clarity, and logical progression in your response. Follow these guidelines:

—

**Guidelines:**
1. **Ensure Comprehensive Coverage**: Identify and include all relevant details visible in the image. Avoid unnecessary repetition or irrelevant information.
2. **Avoid Adding Imaginary Details**: Base your reasoning strictly on what is visible in the image or provided in the description. Do not include fabricated or unverifiable details.
3. **Incorporate Relevant Context**: Add factual, relevant context to enhance understanding where appropriate, but ensure it aligns strictly with the visible or provided content.
4. **Prevent Inaccuracies**: Stick to the given data. Avoid assumptions or deviations from the available evidence.

—

**Step-by-Step Process:**
**Step 1: Extract salient content by identifying the key elements that define the image.**
*Example:* The image is a monochrome photocopy of a document that appears to be a page of meeting or project notes. It contains both typed and handwritten text, with a focus on tasks and progress updates related to paper-related issues. The document includes a reference number at the bottom and a source URL.
**Step 2: Analyze detailed information, focusing on instance-level attributes such as low-level and fine-grained details.**
*Example:* The document lists several tasks, such as checking with "KC" on the possibility of putting bands "long-ways," which is marked as "In progress." Other tasks include checking on "shrinking" paper, which is also "In progress," and checking the commercial viability of banded papers, marked as "Okay." There are handwritten notes and checks next to some points, indicating their status.
**Step 3: Consider relational-level attributes, analyzing interactions between elements and their spatial organization.**
*Example:* The tasks are organized in a list format, with some items having associated handwritten notes that indicate completion or ongoing status. The name "Jimmy Wu" is associated with an action item regarding a DC work request with KC banded papers, awaiting approval for banded additives. The document also mentions running "GPC KS and KOOL KS on RIP-4 (LCC)" and notes that KC is running "cross-hatch" papers.
**Step 4: Examine marginal or peripheral content to ensure no important information is missed.**
*Example:* The document specifies that the next meeting is scheduled for Monday, February 7, at 9:00 a.m. in the International Conference Room. The reference number "584100571" is located at the bottom of the page, and the source URL is included at the bottom.
**Step 5: Organize all observations into a detailed, cohesive description.**
*Example:* The image is a monochrome photocopy of a document that appears to be a page of meeting or project notes, containing both typed and handwritten text. The document lists several tasks related to paper-related issues, such as checking with "KC" on the possibility of putting bands "long-ways," which is marked as "In progress," and checking the commercial viability of banded papers, marked as "Okay." Handwritten notes and checks next to some points indicate their status. The name "Jimmy Wu" is associated with an action item regarding a DC work request with KC banded papers, awaiting approval for banded additives. Other items include running "GPC KS and KOOL KS on RIP-4 (LCC)" and KC running "cross-hatch" papers. The next meeting is scheduled for Monday, February 7, at 9:00 a.m. in the International Conference Room. The document is marked with a reference number "584100571" at the bottom, and a source URL is included.

—

**Important Notes:**
- **Steps 1–4**: Write concise observations in one or two sentences each.
- **Step 5**: Summarize all observations into a detailed paragraph or two, as descriptive as necessary.
**Input:** |<image>| Question: Could you please transcribe the image into a descriptive paragraph? Explain your description step-by-step. Original description: |<caption>|
**Output:**

---

Table 21: The prompt utilized by GPT-4o for evaluating the quality of re-captioned data.

---

**Prompt Used by GPT-4o to Evaluate Image Caption Quality**

---

You are an expert AI assistant tasked with evaluating the quality of captions for images. Your job is to assess the caption's quality based on specific criteria and provide a clear, concise critique followed by structured evaluation scores. Ensure your response follows the exact format below and adheres to the evaluation criteria.

—

**Evaluation Process:**
1. **Critique First**: Begin by generating a concise critique of the caption. Highlight both its strengths and weaknesses in plain language. Focus on how well the caption describes the image and aligns with the criteria.
2. **Score Each Criterion**: After the critique, provide a score for each evaluation criterion on a scale from 1 to 5. Ensure the scores are consistent with the critique and avoid contradictions.

—

**Evaluation Criteria:**
Evaluate the caption based on the following eight dimensions:
1. **Salient Content**: Does the caption highlight the key elements and most important details of the image?
2. **Fine-Grained Details**: Does the caption include specific attributes, such as textures, colors, or text found in the image?
3. **Relational Attributes**: Does the caption describe interactions or spatial relationships between elements in the image?
4. **Peripheral Content**: Does the caption include additional relevant details that enhance completeness without being redundant?
5. **Faithfulness**: Does the caption accurately describe what is visible in the image without adding imaginary or false information?
6. **World Knowledge**: Does the caption incorporate relevant world knowledge, such as context or implied meaning, to enhance its coherence?

—

**Scoring Rubric:**
- **Poor (1)**: Fails to meet the criterion.
- **Fair (2)**: Partially meets the criterion but has noticeable shortcomings.
- **Average (3)**: Adequately meets the criterion but lacks depth or sophistication.
- **Good (4)**: Strongly aligns with the criterion and demonstrates nuanced understanding.
- **Excellent (5)**: Perfectly aligns with the criterion with high-quality descriptions.

—

**Output Format:** Follow this structured format exactly:
1. **Critique**: Write a concise critique (2-4 sentences), summarizing the strengths and weaknesses.
2. **Scores**: Provide a score for each dimension using the following format:
- Salient Content: Score = [[Your Score]]
- Fine-Grained Details: Score = [[Your Score]]
- Relational Attributes: Score = [[Your Score]]
- Peripheral Content: Score = [[Your Score]]
- Faithfulness: Score = [[Your Score]]
- World Knowledge: Score = [[Your Score]]

—

**Important Notes:**
- Focus solely on the caption's alignment with the criteria and the image content.
- Be objective, impartial, and avoid introducing biases.
- Do not let the caption's length, style, or structure influence your evaluation unless it directly affects quality.
- Ensure your response is clear, concise, and follows the exact output format.
**Input:** |<image>| Caption: |<caption>|
**Output:**

---

# R    TRAINING EXAMPLES

Table 22: An example from the image captioning training dataset curated to elicit systematic multimodal perception.

---

**Image Captioning Data for Enhancing Systematic Multimodal Perception**

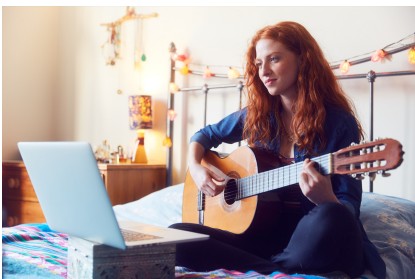

---

**Example with Detailed Description Caption Format**

**Prompt:**
Please provide a vivid description of the image.
**Caption:**
The image features a person with long, red hair sitting cross-legged on a bed, playing an acoustic guitar. They are wearing a dark blue shirt with rolled-up sleeves and black pants, appearing focused on the instrument. The guitar is a classic wooden brown with light-colored frets. In front of them, an open laptop on a small table or stand suggests they might be following music or a tutorial. The room exudes a warm ambiance with soft lighting, and the colorful bedspread adds vibrancy. Decorations like string lights and wall hangings contribute to a bohemian aesthetic, while a dresser with various personal items in the background adds a personal touch to the space.

---

**Example with our proposed *Chain-of-Description* Caption Format**

**Prompt:**
Please provide a vivid description of the image. Describe the image step-by-step.
**Caption:**
**Step 1: Extract salient content by identifying the key elements that define the image.**
The image depicts a person with long, red hair sitting cross-legged on a bed, playing an acoustic guitar. The person is dressed in a dark blue shirt with rolled-up sleeves and black pants. An open laptop is positioned on a small table or stand in front of them, suggesting they might be using it for guidance, such as following music sheets or a tutorial.
**Step 2: Analyze detailed information, focusing on instance-level attributes such as low-level and fine-grained details.**
The guitar is a classic wooden brown with light-colored frets. The room has a warm ambiance, enhanced by soft lighting. The bedspread is colorful, adding vibrancy to the scene. Decorations such as string lights and wall hangings contribute to a bohemian aesthetic. In the background, there is a dresser with various personal items, adding to the lived-in feel of the space.
**Step 3: Consider relational-level attributes, analyzing interactions between elements and their spatial organization.**
The person is seated on the bed, with the guitar resting on their lap, indicating engagement with the instrument. The laptop's placement suggests it is being actively used, possibly for learning or playing along with music. The string lights and wall hangings are positioned to create a cozy and artistic environment.
**Step 4: Examine marginal or peripheral content to ensure no important information is missed.**
The room's decor, including the string lights and wall hangings, adds a personal touch and suggests a creative or artistic personality. The dresser in the background, with its assortment of items, further personalizes the space.
**Step 5: Organize all observations into a detailed, cohesive description.**
The image features a person with long, red hair sitting cross-legged on a bed, playing an acoustic guitar. They are wearing a dark blue shirt with rolled-up sleeves and black pants, appearing focused on the instrument. The guitar is a classic wooden brown with light-colored frets. In front of them, an open laptop on a small table or stand suggests they might be following music or a tutorial. The room exudes a warm ambiance with soft lighting, and the colorful bedspread adds vibrancy. Decorations like string lights and wall hangings contribute to a bohemian aesthetic, while a dresser with various personal items in the background adds a personal touch to the space.

---

Table 23: An example from the visual instruction-following training dataset designed to enhance systematic multimodal reasoning.

---

**Visual Instruction-Following Data for Enhancing Systematic Multimodal Reasoning**

---

**Example 1: Open-Ended Problem**

---

**Example with Direct Answer Response Format**

**Question:**
In the right triangle $ABC$, $\angle C = 90°$, point $D$ is a point on $AC$, $DE \perp AB$ at point $E$, $AB = 10$, $BC = 6$, $DE = 2.4$, then the length of $AD$ is ( )
  A. 1.2   B. 3   C. 4   D. 5

**Response:**
C

---

**Example with Chain-of-Thought Response Format**

**Question:**
In the right triangle $ABC$, $\angle C = 90°$, point $D$ is a point on $AC$, $DE \perp AB$ at point $E$, $AB = 10$, $BC = 6$, $DE = 2.4$, then the length of $AD$ is ( )
  A. 1.2   B. 3   C. 4   D. 5
Answer the question step by step.

**Response:**
**Step 1: Clarify the task objective.**
To solve this problem, I will use the properties of similar triangles and the known measurements provided. By identifying similar triangles, I can set up a proportion to find the length of AD.

**Step 2: Extract the crucial visual information from the image.**
The image shows right triangle $ABC$ with $\angle C = 90°$. Point $D$ is on segment $AC$, and $DE$ is perpendicular to $AB$ at point $E$. Measurements are given as $AB = 10$, $BC = 6$, and $DE = 2.4$.

**Step 3: Generate detailed reasoning to solve the task.**
In the right triangle $ABC$, $\angle C = 90°$. $DE$ is perpendicular to $AB$, making $\angle AED$ also equal to $90°$. Because both triangles $ADE$ and $ABC$ share angle $A$, triangle $ADE$ is similar to triangle $ABC$ by AA similarity (Angle-Angle). Using the similarity of triangles $ADE$ and $ABC$, we can set up the proportion:

$$\frac{AD}{DE} = \frac{AB}{BC}$$

Substituting the known values:

$$\frac{AD}{2.4} = \frac{10}{6}$$

Simplifying the right side:

$$\frac{10}{6} = \frac{5}{3}$$

Thus, the proportion becomes:

$$\frac{AD}{2.4} = \frac{5}{3}$$

Solving for $AD$:

$$AD = \frac{5}{3} \times 2.4 = 4$$

**Step 4: Conclude the task with an answer.**
C

---

Table 24: An example from the visual instruction-following training dataset designed to enhance systematic multimodal reasoning.

---

**Visual Instruction-Following Data for Enhancing Systematic Multimodal Reasoning**

**Example 2: Multiple-Choice Problem**

---

**Example with Direct Answer Response Format**

**Question:**
Why is the person in the air?
  A. Grabbing a frisbee    B. Falling    C. Angry    D. Bouncing
**Response:**
A

---

**Example with Chain-of-Thought Response Format**

**Question:**
Why is he in the air?
  A. grab frisbee    B. falling    C. angry    D. bouncing
Answer the question step by step.
**Response:**
**Step 1: Clarify the task objective.**
I will analyze the image to determine why the person is in the air by examining relevant visual clues. I'll then follow a reasoning process to arrive at the correct answer.
**Step 2: Extract the crucial visual information from the image.**
The image shows an indoor sports setting with a person in mid-air holding an object that resembles a frisbee. The individual is wearing athletic gear, and there is a goal visible in the background, suggesting a sports activity.
**Step 3: Generate detailed reasoning to solve the task.**
Considering the context and the object in hand, it appears the person is actively participating in a sport involving a frisbee. The position of the person suggests they might be jumping to grab or throw the frisbee. The presence of a goal could imply that the activity requires throwing the frisbee accurately, which aligns with the action of jumping.
**Step 4: Conclude the task with an answer.**
A

---

## S    LIMITATIONS

While our experimental results are promising, we consider these findings preliminary, as many aspects of this field remain unexplored.

**Dealing with Trade-Offs between Structured and Unstructured Data.**    We currently adopt two structured methodologies to elicit high-quality multimodal data. While effective, these approaches may serve as a "temporary shortcut" for enhancing model capabilities (Wei & Chung, 2024). To encourage "describing or (implicit) thinking freely," we also incorporate a mix of detailed descriptions and direct answers. Future work may further explore the trade-offs between structured and unstructured data.

**Balancing Data Ratios and Formats in Self-Generated Pre-Training Data.**    Our study primarily aims to establish a starting point for building next-generation foundation MLLMs through a fully self-improving paradigm. As such, we did not focus on optimizing the balance of data types or formats within the self-generated pre-training corpus. Nevertheless, Appendix G shows that varying the proportions of caption data, visual instruction tuning data, and text-only prompts significantly impacts SICoG's performance. Furthermore, Table 14 demonstrates that even when using the same CoD data, different formatting yields different results. Future work may investigate how balancing data ratios and formats can further optimize self-improvement.

**Leveraging More Advanced Quality Evaluation Methods.**    We employ a simple but effective self-consistency mechanism to select high-quality outputs for unlabeled images, image–question pairs, and text-only prompts, based on semantic coherence. Future work may benefit from incorporating more advanced quality evaluation methods to further enhance data selection.

