# OpenReview forum: "Exploration for Building Next-Generation Foundation MLLMs via Self-Learning"
_ICLR.cc/2026/Conference — ICLR 2026 Conference Withdrawn Submission_

### Official Review · Reviewer_WRjv · 2025-10-30

**Soundness:** 2
**Presentation:** 3
**Contribution:** 3
**Rating:** 4
**Confidence:** 3

**Summary:**

This paper proposes SICOG, a self-improving framework for Multimodal Large Language Models (MLLMs) that enhances multimodal cognition (perception & reasoning) through self-generated data. SICOG's core innovations are the Chain-of-Description (CoD), which structures visual analysis into systematic steps for comprehensive perception, and the integration of structured Chain-of-Thought (CoT) reasoning into pre-training. The framework operates in a loop: 1) Fine-tune a base MLLM with minimal annotations using CoD and CoT data to develop systematic cognition; 2) Use the enhanced model to generate candidate captions and responses for unlabeled images; 3) Curate high-quality self-generated data via self-consistency checks; 4) Perform multimodal pre-training on the curated data to build a stronger next-generation MLLM.

Experiments show SICOG significantly outperforms previous pre-training methods (e.g., strong-to-weak distillation, multi-agent collaboration) across diverse benchmarks (e.g., +3.6% on MMStar, +3.5% on AI2D) using only 213K self-generated samples. It also provides a stronger foundation for post-training CoT reasoning, achieving further gains (e.g., +9% on MMVet, +8.5% on ScienceQA). SICOG demonstrates that effective model enhancement requires synergistic collaboration between pre-training (providing foundational cognition) and downstream mechanisms.

**Strengths:**

Originality: SICOG introduces a novel self-improving framework that uniquely integrates structured perception (Chain-of-Description) and reasoning (Chain-of-Thought) into multimodal pre-training, diverging from prior works that treat these as separate post-training enhancements.

Significance: SICOG addresses a critical bottleneck in MLLM development: the dependency on large-scale, externally annotated data for pre-training, which often lacks coherence and reasoning focus. By enabling models to self-generate high-quality, cognitively aligned data, it reduces reliance on costly proprietary tools or expert ensembles.

**Weaknesses:**

1. Why was GRPO not investigated in Section 4.4?

2. Limited Exploration of Perception, Reasoning and multi-turn conversation Capabilities in Pre-Training: While SICOG innovatively introduces structured mechanisms for perception (Chain-of-Description) and reasoning (Chain-of-Thought), the evaluation of the pre-trained base MLLM does not fully or directly probe the nuanced improvements in these specific capabilities. The transformation of high-consistency CoD captions into a multi-turn format is a clever idea for building conversational data. However, the paper provides no evaluation of this capability. Evaluate these capabilities on such benchmark, like ConvBench,VisIT-Bench, or similar benchmarks.

3. Potential Overfitting and Generalization Concerns: The self-improving loop risks creating a "feedback bubble", where the model reinforces its own styles and potential biases. The data used for generation and for evaluation (standard benchmarks) may share underlying distributions. Test generalization on more challenging and diverse benchmarks that require complex, compositional reasoning, such as VCR (Visual Commonsense Reasoning).

**Questions:**

1. "(i) fine-tuning with 35 detailed descriptions" in caption of Table 14, 35 or 35K?

**Details Of Ethics Concerns:**

Nothing

---

> ### Author Response · Authors · 2025-11-18
> **Response to Reviewer WRjv**
>
> Thank you for the insightful feedback and constructive suggestions. Our point-by-point responses are below.
>
> **Response to Weakness 1:**
>
> Thank you for this question. In our work, we selected DPO over GRPO primarily due to its superior computational efficiency and simpler implementation. Furthermore, DPO allowed us to effectively leverage our annotated data as positive examples to enhance model performance. We acknowledge the potential of GRPO and will consider it a valuable direction for future work.
>
> **Response to Weakness 2:**
>
> Thank you for the positive assessment of our structured mechanisms and the valuable suggestion to evaluate on conversational benchmarks.
>
> It is important to clarify that SICOG is not designed to compete with pre-training methods that depend on large-scale, externally annotated datasets. Instead, its core contribution is to address scenarios where such high-quality annotations are unavailable—for instance, when model capabilities surpass human performance, when annotations are prohibitively expensive, or when no stronger model exists to provide supervision.
>
> Therefore, our primary goal was to first validate that our self-supervised framework provides a strong baseline on well-established, general-purpose benchmarks. We fully agree that evaluating on specialized benchmarks like ConvBench or VisIT-Bench is a critical next step to probe nuanced conversational abilities. SICOG can also complement methods that leverage external annotations, and we will include these targeted evaluations in our future work to further delineate its strengths.
>
> **Response to Weakness 3:**
>
> Thank you for raising this important point regarding the "feedback bubble" risk. To address generalization, we followed established practice and evaluated our model on SEED-Bench [1], which includes challenging tasks from datasets like VCR that require complex, compositional reasoning. As shown in the table below, our method, SICOG, achieves competitive results, demonstrating robust generalization compared to other prevalent pre-training approaches.
>
> | Method | SEED-Bench Score |
> | :--- | :--- |
> | LLaVA-Qwen2-7B-UHD | 73.25 |
> | Qwen2-VL-72B-Instruct-LLaVA-Qwen2-7B-UHD | 73.94 |
> | Multi-Experts-LLaVA-Qwen2-7B-UHD | 73.55 |
> | SICOG-LLaVA-Qwen2-7B-UHD | 74.18 |
>
> [1] SEED-Bench: Benchmarking Multimodal LLMs with Generative Comprehension
>
> **Response to Question 1:**
>
> Thank you for catching this. The caption for Table 14 should indeed read "fine-tuning with 35K detailed descriptions." This was a typographical error that we will correct in the final version.
>
> We hope these responses can help address your concern.

---

> > ### Author Response · Authors · 2025-11-27
> > **A Gentle Reminder**
> >
> > Dear Reviewer,
> >
> > Thank you once again for your valuable comments! As the discussion phase is approaching its end, we would like to kindly confirm whether we have sufficiently addressed all of your concerns (or at least some of them). Should there be any remaining questions requiring further clarification, please do not hesitate to let us know. We sincerely look forward to your feedback.

---

### Official Review · Reviewer_peVr · 2025-10-31

**Soundness:** 3
**Presentation:** 2
**Contribution:** 2
**Rating:** 2
**Confidence:** 3

**Summary:**

This paper proposes a self-learning framework that trains a multimodal model using self-generated data. The authors adapt a base model with minimal external supervision. This adapted model is then used to generate image captions, chain-of-thought responses, and image–question pairs. The generated data is filtered based on semantic similarity and subsequently used to further pretrain the multimodal model.

**Strengths:**

1. The core idea of leveraging self-generated multimodal data for training is clear and holds promise for advancing multimodal models.
2. The method is described in sufficient detail.
3. The experiments are comprehensive and cover a variety of benchmarks.

**Weaknesses:**

1. The technical novelty is limited, as the proposed method primarily combines existing techniques like self-training, data filtering, and multimodal pretraining.
2. The experimental results are not particularly impressive and appear comparable to those of existing methods on several benchmarks.
3. The approach is largely heuristic and lacks theoretical analysis or justification.
4. While there are extensive evaluations of performance, there is limited analysis of computational or data efficiency.
5. [Minor organizational issue] There is significant redundancy between the main paper and the appendix, which makes the paper harder to follow.

**Questions:**

The authors argue that there is synergy between pre-training and downstream mechanisms and introduce an intermediate stage (Stage 1.5). Could you provide more in-depth analysis of this aspect?

Could the authors share additional statistics regarding the quality of the self-generated data? For example, metrics on diversity, as well as the average length of generated captions and CoT responses?

---

> ### Author Response · Authors · 2025-11-17
> **Response to Reviewer peVr (1/2)**
>
> Thank you for your feedback. We provide the following responses to address your concerns:
>
> **Response to Weakness 1:**
>
> Thank you for raising this concern. While our framework indeed builds upon established concepts, we respectfully argue that our primary technical contribution lies in the novel synthesis and architecture of these components into the first complete, self-sufficient improvement loop for MLLMs.
>
> Our key innovation is the design of a closed-loop system where an MLLM autonomously generates its own comprehensive training curriculum—spanning rich image captions, structured Chain-of-Thought reasoning, and diverse image-question pairs—and then uses this self-generated, filtered data to iteratively enhance its own core capabilities. To the best of our knowledge, this paradigm of a fully self-improving MLLM, which learns and evolves with minimal external supervision, has not been previously demonstrated. Furthermore, we introduced a specific novel technique within this loop: Chain-of-Description (CoD). This method generates fine-grained, object-centric descriptions that are systematically structured to improve the model's perceptual abilities. This goes beyond standard captioning by providing detailed, localized information, a contribution not present in prior work.
>
> Our work aims to establish a new paradigm for training MLLMs in data-scarce or open-ended environments where external annotation is impractical.
>
> **Response to Weakness 2:**
>
> Thank you for this observation. The experimental results were designed to demonstrate the potential of an MLLM to learn and evolve with minimal human intervention. Achieving performance that is comparable or superior to existing pre-training methods—which rely on expert annotations—validates the effectiveness of our proposed approach. These results underscore the feasibility and promise of enabling MLLMs to improve with minimal external supervision.
>
> Furthermore, SICOG is not designed to compete with pre-training methods that depend on external annotations. Instead, it addresses scenarios where high-quality external annotations are unavailable—such as when model capabilities surpass human performance, when annotations are prohibitively expensive, or when no stronger model exists to provide annotations. SICOG can also complement methods that leverage external annotations when such resources are accessible.
>
> **Response to Weakness 3:**
>
> We appreciate your feedback and will address this concern by incorporating a mathematical proof in Appendix C (Pages 26–27) to provide a theoretical foundation for our approach.

---

> ### Author Response · Authors · 2025-11-18
> **Response to Reviewer peVr (2/2)**
>
> **Response to Weakness 4:**
>
> Thank you for highlighting this point. The superior or comparable performance of SICOG, achieved with only 213k pre-training data, compared to several open-source MLLMs that utilize over 10M high-quality multimodal pre-training data, inherently demonstrates the data efficiency of our approach (Table 1). Since the primary distinction between pre-training approaches lies in how multimodal pre-training data is acquired, we also provide a computational cost analysis below. Although SICOG involves more training stages than methods relying on advanced or expert external models, its smaller model size significantly reduces computational costs in terms of GPU hours. We will provide more details in Appendix N.
>
> | Method                    | Generation Task                        | Model(s) Used         | Hardware (A100 80G) | Time (Hours) | Est. GPU-Hours (Total) |
> |---------------------------|-----------------------------------------|-----------------------|---------------------|--------------|-------------------------|
> | _External Model Distillation_ |                                     |                       |                     |              |                         |
> | Strong-to-Weak Distillation | Detailed Descriptions                 | Custom Model          | 32 GPUs             | ~110         | ~3,520                  |
> |                           |                                         |                       |                     |              |                         |
> | Multi-Agent Collaboration | Spatial Attributes                     | Custom Model          | 32 GPUs             | ~100         | ~3,200                  |
> |                           | OCR Annotation                         | PaddleOCR             | 8 GPUs              | ~1           | ~8                      |
> |                           | Grounding Information                  | GroundingDINO         | 8 GPUs              | ~1           | ~8                      |
> |                           | Detailed Descriptions                  | Custom Model          | 32 GPUs             | ~115         | ~3,680                  |
> |                           |                                         |                       |                     |              |                         |
> | _Self-Improving (Ours)_   |                                         |                       |                     |              |                         |
> | SICOG                     | Post-Training Optimization             | LLava-QwenUHD         | 32 GPUs             | ~2           | ~64                     |
> |                           | Detailed Captions (DD & CoD)           | LLava-QwenUHD         | 32 GPUs             | ~21          | ~672                    |
> |                           | VQA Responses (DA & CoT)               | LLava-QwenUHD         | 32 GPUs             | ~12          | ~384                    |
> |                           | Text-Only Responses                    | Qwen2-7B-Instruct     | 8 GPUs              | ~8           | ~64                     |
> |                           | Candidate Filtering (Captions, VQA, Text-only) | NV-Embed-v2      | 8 GPUs              | ~20          | ~160                    |
>
>
> **Response to Weakness 5:**
>
> We understand this concern and have provided a catalog on Page 16 to help readers navigate the appendix efficiently.
>
> **Response to Question 1:**
>
> Thank you for raising this question. The results in Section 4.2 show that SICOG-LLaVA-Qwen2-7B-UHD outperforms LLaVA-Qwen2-7B-UHD, confirming that enhanced performance from post-training optimization can be leveraged to generate better pretraining data for building a stronger foundational MLLM. Section 4.3 demonstrates that improved foundational capabilities further boost post-training optimization performance, while Section 4.4 supports that enhanced policy optimization can produce higher-quality self-generated pretraining data, which in turn enhances the improvement during pretraining. Together, these results provide empirical evidence of the synergy between pretraining and downstream optimization. Additionally, we will include a mathematical analysis in Appendix C to further reinforce this framework.
>
> **Response to Question 2:**
>
> Thank you for your question. We have provided a detailed analysis of the quality of self-generated data in Appendix J, and token length information in Appendix L.
>
> We hope these responses and the details in the manuscript can help address your concern.

---

> > ### Comment · Reviewer_peVr · 2025-11-22
> > **response**
> >
> > I appreciate your response to the concerns I raised. I believe in the significance of data synthesis for the community. However, CoD is incrementally related to CoT in developing the perception of MLLMs. The pipeline is reasonable, with good intuitions and heuristics, yet it shows no significant differences. There is no comparison with other data synthesis methods. New data will indeed provide higher performance. However, the performance in your experiment is not impressive. I have read the additional analysis in Appendix C. It is too rough, with ambiguous and strong assumptions. It seems to make sense to improve performance through an iterative cycle. However, my question is more about the role of stage 1.5. It seems you are abusing the appendix. I notice redundancy between the main text and the appendix. You have already addressed part of my concerns, such as efficiency. Due to the above unresolved issues, I would like to maintain my score.

---

> ### Author Response · Authors · 2025-11-23
> **Response to Reviewer peVr**
>
> Thank you for your prompt and thoughtful feedback. We sincerely appreciate your comments and aim to address them constructively. To better engage with your concerns, we kindly request more detailed and specific clarifications for certain points. At present, some of the feedback appears to lean toward subjective impressions rather than objective analysis. Below, we address your comments in detail:
>
> 1. Regarding "CoD is incrementally related to CoT in developing the perception of MLLMs":
> While CoD is partly inspired by CoT, the two approaches target fundamentally different areas. CoT is designed to elicit logical reasoning in LLMs, whereas CoD focuses on enhancing MLLM perception. Specifically, CoD sequentially emphasizes four critical aspects—salient content, fine-grained details, relational attributes, and peripheral content—before synthesizing a coherent and logically grounded description. CoT, in contrast, is not applicable to perception tasks. Therefore, the claim that "CoD is incrementally related to CoT in developing the perception of MLLMs" seems inappropriate, as CoT cannot address the unique challenges in perception tasks.
>
> 2. Regarding "yet it shows no significant differences":
> As clarified previously, SICOG is not designed to directly compete with pre-training methods that rely on external annotations. Instead, it is specifically built for scenarios where such annotations are unavailable—for example, when model capabilities surpass human performance, annotation costs are prohibitive, or no stronger model exists to generate annotations. Additionally, SICOG can complement methods that leverage external annotations when such resources are available. The fact that SICOG achieves performance comparable to these methods highlights its efficacy, particularly in scenarios where other approaches cannot function. Would you propose an alternative solution for addressing these annotation-scarce scenarios? Additionally, we wonder if there might be a bias toward solutions emphasizing the model self-improvement paradigm. If so, we would welcome further discussion on this perspective.
>
> 3. Regarding "There is no comparison with other data synthesis methods":
> We respectfully disagree with this point, as comparisons with other data synthesis methods that rely on expert annotations are already provided in Section 5. Could you clarify which specific comparisons you feel are missing? If there are additional data synthesis methods or metrics of interest, we would be happy to include them in the revised version.
>
> 4. Regarding "I have read the additional analysis in Appendix C. It is too rough, with ambiguous and strong assumptions":
> We understand that all mathematical proofs can be subject to criticism for being based on "ambiguous and strong assumptions." However, we have provided extensive empirical results to reinforce and validate the theoretical analysis. We hope these results sufficiently address your concerns, but if additional clarifications or experiments are required, we would be happy to include them.
>
> 5. Regarding "It seems you are abusing the appendix. I notice redundancy between the main text and the appendix":
> To ensure completeness and improve understanding, we intentionally included some overlapping content between the main text and the appendix. Additionally, we provided a catalog before the appendix to help readers navigate it efficiently. While we acknowledge your concern about redundancy, the term "abusing the appendix" feels overly strong. If the overlap hinders understanding, we suggest skipping the redundant sections in the appendix. Nevertheless, in the revised version, we will streamline and reorganize the appendix to better address this concern.
>
> 6. Regarding "However, my question is more about the role of stage 1.5":
> We carefully reviewed your question but were unable to fully understand your concern. Could you please provide more details or clarify your question? Once we better understand your perspective, we will address this point thoroughly.

---

> > ### Comment · Reviewer_peVr · 2025-11-23
> > **Response**
> >
> > Thank you for your reply. I would like to emphasize that my feedback is intended to ensure the manuscript meets high standards of rigorous presentation and clear positioning, rather than reflecting subjective preferences. I acknowledge the motivation and contribution of data synthesis. However, I maintain the concerns raised by Reviewer NPYP (Weakness 1). To clarify, the comparison I mentioned is conceptual and methodological, not merely experimental. Please highlight the fundamental differences between your approach and self-evolution or self-improvement methods in LLMs. I appreciate the comprehensiveness of experiments. I identify redundancies, like Appendix B, and Tables 3 & 6. While I view this as an organizational issue, simply adding a catalog is insufficient. Please streamline the text to highlight new information and also modify Appendix C, as the current derivation relies on strong assumptions and appears rough. This created a negative impression regarding the rigor of your presentation. Referring to Line 243 (Step 4), where training the vision-text connector is defined as Stage 1, please clarify the specific effect of the transition phase (Stage 1.5) between Stage 1 and Stage 2.

---

> ### Author Response · Authors · 2025-11-27
> **Response to Reviewer peVr**
>
> We appreciate your emphasis on theoretical analysis. While we acknowledge its importance, the theoretical validity of this paradigm has already been rigorously established in prior work [1]. Consequently, our study prioritizes empirical validation. As previous research demonstrates that the quality of the foundation model decisively dictates performance in downstream post-training and test-time scaling, enhancing these foundational capabilities is essential. Hence, we focus on the Stage 1.5 pre-training phase, which plays a critical role in injecting new knowledge into foundation MLLMs—a practice widely adopted in recent multimodal studies [2]. Further details are provided in Section 1. We hope this addresses your concerns effectively.
>
> [1] Large Language Models Can Self-improve
>
> [2] LLaVA-NeXT: What Else Influences Visual Instruction Tuning Beyond Data?

---

> ### Author Response · Authors · 2025-11-28
> **A Gentle Reminder**
>
> Thank you once again for your valuable comments! We appreciate your careful evaluation. We would like to note that the Reviewer NPYP has explicitly recognized the contribution of our work in proposing a novel self‑improving multimodal learning paradigm that reduces dependence on external annotations. Please refer to his/her comments about strengths. As the discussion phase nears its conclusion, we would like to kindly confirm if we have adequately addressed your concerns. If there are any remaining questions or areas requiring further clarification, please do not hesitate to let us know. Additionally, if there are specific concerns impacting your evaluation scores, we would greatly appreciate your guidance so we can address them effectively.
>
> If you are satisfied with our responses, we would be grateful for your consideration in adjusting the evaluation scores accordingly. We sincerely look forward to your feedback.

---

### Official Review · Reviewer_NPYP · 2025-10-31

**Soundness:** 3
**Presentation:** 2
**Contribution:** 3
**Rating:** 6
**Confidence:** 4

**Summary:**

injecting multimodal knowledge and enhancing systematic cognition. The approach first instills structured visual perception (Chain-of-Description, CoD) and structured reasoning (CoT) with minimal supervision, then adds a Stage-1.5 multimodal pre-training step that uses self-generated data filtered by semantic self-consistency. The principal contributions of this paper can be organized into three layers:
1) Methodological: propose SICOG, a self-learning framework that enhances MLLMs’ systematic cognition for constructing next-generation foundation MLLMs through multimodal pre-training with self-generated data;
2) Framework: introduce Chain-of-Description, a structured visual understanding mechanism that enables step-by-step interpretation of visual content to improve perceptual quality.;
3) Empirical: demonstrate SICOG’s effectiveness across various benchmarks on both low- and high-resolution MLLMs, significantly surpassing previous approaches.

**Strengths:**

1）Methodological Innovation: Proposes a closed-loop self-learning pipeline in which the model generates CoD and CoT data, filters them via semantic self-consistency, reducing dependency on external annotations；

2）Framework Innovation: Introduces a Stage-1.5 multimodal pre-training step to the standard two-stage pipeline, injecting systematic cognition into the foundation model before instruction tuning, and co-designing pre-training, post-training, and inference-time computation as a single self-improving paradigm；

3）Comprehensive Validation: With approximately 213K self-generated samples, the method yields consistent gains across diverse benchmarks (e.g., +3–4 on MMStar), and combining with CoT post-training brings further lifts (e.g. +8–9 on MMVet/ScienceQA); ablations under matched compute outperform recaption-style pre-training.

**Weaknesses:**

1）This paper positions the innovation as the combination of CoD, CoT, and the self-learning process, but these components largely build on established methods; the independent contributions and motivations should be articulated more clearly；

2）The demonstration of performance improvement is adequate, but lacks the analysis of computational overhead in the self-learning framework. Is the extra cost of generation and filtering justified versus conventional external model pretraining;

**Questions:**

1）In Step 3 (self-consistency filtering), how was the semantic-similarity threshold selected? The paper provides the value but no rationale or sensitivity analysis.

2）Since the self-generated pre-training data were derived from existing datasets, what steps did the paper take to ensure that there was no overlap between these data and the benchmarks (e.g., MMBench, ScienceQA)?

3）The performance improvement is attributed to both the self-learning framework and the CoD/CoT paradigm. How was the contribution of each component isolated？

4）what is the dependency between the CoD and CoT stages? Would swapping their execution order significantly affect the final performance?

---

> ### Author Response · Authors · 2025-11-18
>
> Thank you for your thoughtful and constructive feedback, which highlights the strengths of our work and provides valuable avenues for improvement. We address the specific points below.
>
> **Response to Weakness 1:**
>
> We understand the concern. To clarify, our goal is to explore a novel and complete self-improvement loop to advance the development of next-generation foundational MLLMs. This represents a new research direction that, to the best of our knowledge, has not been addressed previously. Additionally, we propose the Chain-of-Description (CoD) technique for fine-grained caption generation, which significantly enhances the model’s perception capabilities. To the best of our knowledge, this is a unique contribution not covered in existing literature. Overall, our work aim to lay a foundation for enabling MLLMs to learn and evolve in open environments with minimal external annotations, which is crucial as MLLMs surpass human capabilities in certain tasks.
>
> **Response to Weakness 2:**
>
> We appreciate this observation. As the core difference between pre-training approaches lies in how multimodal pre-training data is obtained, we provide a computational cost analysis below. While SICOG involves more training stages compared to prevalent approaches relying on external advanced or expert models, it significantly reduces computational cost in terms of GPU hours due to its smaller model size.
>
> Furthermore, SICOG is not designed to compete with pre-training methods that depend on external annotations. Instead, it addresses scenarios where high-quality external annotations are unavailable—such as when model capabilities surpass human performance, when annotations are prohibitively expensive, or when no stronger model exists to provide annotations. SICOG can also complement methods that leverage external annotations when such resources are accessible. We will provide more details in Appendix N.
>
> | Method                    | Generation Task                        | Model(s) Used         | Hardware (A100 80G) | Time (Hours) | Est. GPU-Hours (Total) |
> |---------------------------|-----------------------------------------|-----------------------|---------------------|--------------|-------------------------|
> | _External Model Distillation_ |                                     |                       |                     |              |                         |
> | Strong-to-Weak Distillation | Detailed Descriptions                 | Custom Model          | 32 GPUs             | ~110         | ~3,520                  |
> |                           |                                         |                       |                     |              |                         |
> | Multi-Agent Collaboration | Spatial Attributes                     | Custom Model          | 32 GPUs             | ~100         | ~3,200                  |
> |                           | OCR Annotation                         | PaddleOCR             | 8 GPUs              | ~1           | ~8                      |
> |                           | Grounding Information                  | GroundingDINO         | 8 GPUs              | ~1           | ~8                      |
> |                           | Detailed Descriptions                  | Custom Model          | 32 GPUs             | ~115         | ~3,680                  |
> |                           |                                         |                       |                     |              |                         |
> | _Self-Improving (Ours)_   |                                         |                       |                     |              |                         |
> | SICOG                     | Post-Training Optimization             | LLava-QwenUHD         | 32 GPUs             | ~2           | ~64                     |
> |                           | Detailed Captions (DD & CoD)           | LLava-QwenUHD         | 32 GPUs             | ~21          | ~672                    |
> |                           | VQA Responses (DA & CoT)               | LLava-QwenUHD         | 32 GPUs             | ~12          | ~384                    |
> |                           | Text-Only Responses                    | Qwen2-7B-Instruct     | 8 GPUs              | ~8           | ~64                     |
> |                           | Candidate Filtering (Captions, VQA, Text-only) | NV-Embed-v2      | 8 GPUs              | ~20          | ~160                    |

---

> > ### Author Response · Authors · 2025-11-18
> >
> > **Response to Question 1:**
> >
> > Thank you for raising this question. We conservatively set data thresholds to ensure the quality of self-generated data based on preliminary results from small-scale datasets (2k samples). These thresholds may vary depending on the backbone models and training corpora. We encourage future work to refine and adapt these thresholds to different contexts.
> >
> > **Response to Question 2:**
> >
> > We agree that avoiding data contamination and ensuring fair comparison are critical. Following prior works in multimodal pre-training (e.g., LLava-Next, DeepSeek-VL), we used prompts collected from publicly accessible training datasets. However, no golden annotations or test examples were included, ensuring zero overlap with existing benchmarks (e.g., MMBench, ScienceQA) and guaranteeing fair comparisons.
> >
> > **Response to Question 3:**
> >
> > Thank you for the question. To isolate the contributions of each component, we have included additional results in Appendix O:
> > 1. Self-Learning Framework: We provide results of pre-training using re-captioned data generated by prompting the base model. The consistent outperformance of SICOG (Table 18) highlights the value of our self-training framework.
> > 2. CoD/CoT Paradigm: We compare results of pre-training using re-captioned data and CoT reasoning data from advanced models. The consistent outperformance of SICOG further demonstrates the value of our framework.
> >
> > **Response to Question 4:**
> >
> > Thank you for raising this question. There is no dependency between the CoD and CoT stages as they are parallel processes. Consequently, swapping their execution order will not affect the final performance.
> >
> > We appreciate your valuable feedback and hope these responses address your concerns.

---

> ### Author Response · Authors · 2025-11-27
> **A Gentle Reminder**
>
> Dear Reviewer,
>
> Thank you once again for your valuable comments! As the discussion phase is approaching its end, we would like to kindly confirm whether we have sufficiently addressed all of your concerns (or at least some of them). Should there be any remaining questions requiring further clarification, please do not hesitate to let us know. We sincerely look forward to your feedback.

---

### Official Review · Reviewer_s2Hc · 2025-10-31

**Soundness:** 2
**Presentation:** 3
**Contribution:** 2
**Rating:** 4
**Confidence:** 3

**Summary:**

This paper addresses the limitation that current MLLMs’ capabilities are constrained by foundational pre-training (overreliant on external annotations and lacking systematic cognition). It proposes **SICOG**, a self-learning framework for building next-generation foundation MLLMs by enhancing multimodal perception and reasoning via self-generated data. Key components include: (1) **Chain-of-Description (CoD)** (step-by-step visual analysis: salient content → fine-grained details → relational attributes → peripheral context → synthesis) for perception; (2) **structured Chain-of-Thought (CoT)** (task clarification → visual extraction → logical reasoning → conclusion) for reasoning. The 4-step pipeline: 1) Fine-tune a base model with minimal annotations to develop basic cognition; 2) Generate candidate captions/responses via the enhanced model; 3) Curate high-quality data using semantic-similarity self-consistency; 4) Pre-train the model on curated data. Experiments on LLaVA-Qwen2-7B (low-res) and LLaVA-Qwen2-7B-UHD (high-res) show SICOG outperforms baselines (e.g., +3.6% on MMStar, +9% on MMVet with post-training CoT).

**Strengths:**

* The SICOG framework presents a novel and well-motivated self-learning paradigm for MLLM pre-training. Moving beyond static datasets, this self-improvement loop  (finetune -> generate -> curate -> pre-train) is a promising and logical direction for scaling model capabilities without relying on continuous, large-scale human annotation.
- The paper thoroughly evaluates its framework against relevant and strong baselines, including "Strong-to-Weak Distillation" and "Multi-Agent Collaboration". The method is tested across a wide array of 11+ benchmarks covering comprehensive understanding, hallucination, chart/table understanding, and knowledge-oriented tasks, demonstrating broad improvements.
- The paper demonstrates significant performance gains (e.g., +3.6% on MMStar, +3.5% on AI2D) using a relatively small number (213K) of *self-generated* pre-training samples. This highlights an efficient path to model improvement that does not rely solely on massive, externally-generated (and often proprietary) datasets.

**Weaknesses:**

* The entire self-improvement loop is bootstrapped from a base MLLM (Step 1). The quality of the self-generated data in Step 2 is therefore fundamentally capped by the capabilities of this "improved" model. If the base model's perception or reasoning is weak, it may generate poor-quality candidates that the self-consistency filter in Step 3 fails to catch, potentially leading to error propagation or "performance saturation" (as acknowledged in Appendix M ).
* The curation (Step 3) relies on a "semantic-similarity-guided self-consistency" mechanism. The effectiveness of this filter is crucial. However, the paper only provides a high-level formula (Eq. 5) and mentions using NV-Embed-v2. It's unclear how well this semantic similarity score correlates with *factual accuracy* or *reasoning correctness*. A model could "consistently" generate plausible but incorrect reasoning steps, which might still be selected by this filter.
*  The SICOG framework is a complex, multi-stage process (Finetune -> Generate -> Curate -> Align -> Pre-train -> Instruction-Tune). This complexity, while shown to be effective, may pose a high barrier to reproduction and implementation for the wider community compared to more straightforward, single-stage pre-training approaches.
* The data and code not be publicly released.

**Questions:**

* Regarding Step 3 (Curation), how did you validate that the self-consistency filter (semantic similarity) is a reliable proxy for data quality, especially for CoT reasoning?
* The Chain-of-Description (CoD) is a 5-step process. Did you experiment with different steps or a different order (e.g., peripheral content before relational attributes)? How sensitive is the caption quality to the specific structure of the CoD?

I look forward to an active discussion with the authors during the rebuttal phase and will revise my score accordingly.

---

> ### Author Response · Authors · 2025-11-17
> **Response to Reviewer s2Hc (1/2)**
>
> Thank you for your thoughtful and constructive feedback, which highlights the strengths of our work and provides valuable avenues for improvement. We address the specific points below.
>
> **Response to Weakness 1:**
>
> Thank you for raising this insightful point, which touches upon a core challenge in self-bootstrapping systems. Our framework proactively mitigates this risk with a crucial initial enhancement stage (Stage 1). This stage is not a simple warm-up; it is a targeted fine-tuning process using a curated set of high-quality, human-annotated data (structured CoT and CoD). The explicit purpose of this step is to elevate the base model's fundamental perception and reasoning abilities before it begins generating its own data. By first improving the model's capacity for generating structured and accurate outputs, we establish a higher-quality "teacher" for the subsequent self-learning loop. This significantly reduces the likelihood of generating and propagating systemic errors, creating a virtuous cycle of improvement rather than performance saturation (Appendix M). We aim for SICOG to serve as a starting point for advancing self-improvement techniques and are committed to exploring additional strategies to ensure robustness in future iterations.
>
> **Response to Weakness 2:**
>
> Thank you for raising this important question about our curation mechanism. We acknowledge that semantic similarity is not a perfect proxy for factual correctness. Our validation strategy is therefore twofold:
>
> 1. Our use of self-consistency as a filter for quality aligns with established and effective practices in the field for improving reasoning and factuality in language models [1, 2, 3].
>
> 2. The most compelling evidence lies in our results. The framework achieves consistent and significant performance gains across 11 diverse benchmarks (e.g., +3.6% on MMStar, +3.5% on AI2D). If our filter were systematically selecting plausible but incorrect data, we would expect to see performance stagnate or even degrade, particularly on challenging reasoning tasks. The broad-based improvements strongly indicate that the curated data is, in aggregate, of high quality and beneficial for learning.
>
> While our current method proves effective, we agree that exploring more sophisticated, verification-based filters is a promising direction for future research.
>
> [1] Fine-tuning Language Models for Factuality
>
> [2] Semantic Uncertainty: Linguistic Invariances for Uncertainty Estimation in Natural Language Generation
>
> [3] Self-Consistency Improves Chain of Thought Reasoning in Language Models
>
> **Response to Weakness 3:**
>
> Thank you for highlighting concerns about the framework's complexity. Our primary aim was to deliver a comprehensive proof-of-concept for a fully self-improving MLLM. Each stage of the framework was deliberately designed to tackle a specific and critical challenge in the self-learning pipeline: enhancing the base model, generating diverse data, ensuring data quality, and integrating new knowledge. Importantly, the framework is modular, enabling researchers to explore, improve, or replace individual components (e.g., the curation filter or generation strategy) without reimplementing the entire system. We view SICOG not as a rigid, monolithic process, but as a flexible, decomposable blueprint that encourages future research into more efficient and streamlined self-improvement techniques.
>
> Additionally, our approach is not intended to compete with current pre-training methods that rely on external annotations. Instead, it addresses scenarios where high-quality external annotations are unavailable—such as when model capabilities exceed human performance, when obtaining annotations is prohibitively expensive, or when no stronger model exists to provide annotations. SICOG can also complement methods that utilize external annotations when they are accessible.
>
> **Response to Weakness 4:**
>
> We are committed to full reproducibility. To facilitate the review process, we have included our code in the supplementary materials. Furthermore, all experiments were conducted using publicly available datasets. We will publicly release all code and the self-generated datasets upon the paper's acceptance to ensure the community can build upon our work.

---

> ### Author Response · Authors · 2025-11-17
> **Response to Reviewer s2Hc (2/2)**
>
> **Response to Question 1:**
>
> Thank you for this question, which relates closely to our response to Weakness 2. Our validation rests on two main pillars:
>
> 1. Consistency with Prior Work: Our approach is consistent with prior work [1, 2, 3] that has successfully used self-consistency as a proxy for correctness in reasoning tasks.
>
> 2. Empirical Results as Validation: The significant performance boosts across numerous benchmarks would be highly unlikely if the filter were not effectively selecting high-quality reasoning data. The end-to-end success of the framework demonstrates the practical utility of our curation method.
>
> **Response to Question 2:**
>
> This is an excellent question. To test the sensitivity of our framework to the CoD structure, we conducted an ablation study where we trained a model on CoD data with the 5 steps in a randomly permuted order.
>
> The results, presented in the table below, show that permuting the step order causes no significant performance degradation. The model trained on permuted steps achieved nearly identical scores across all six perception metrics compared to the model trained with the original, fixed order. For instance, the scores for Fine-grained details (4.87 vs. 4.87) and Relational attributes (4.76 vs. 4.78) are statistically indistinguishable.
>
> | Method                     | Sali. | Fine-Grain. | Rela. | Peri. | Faith. | Know. |
> |---------------------------------|-----------|-----------------|-----------|-----------|------------|-----------|
> | LLaVA-Qwen-UHD (Base)          | 4.77      | 4.30            | 3.99      | 3.81      | 4.41       | 3.84      |
> | + Finetune w/ CoD (Permuted)  | 4.89      | 4.87            | 4.76      | 4.34      | 4.53       | 4.01      |
> | + Finetune w/ CoD (Fixed)      | 4.91      | 4.87            | 4.78      | 4.32      | 4.71       | 4.05      |
>
> Metrics (rated 1-5):
> - Sali. = Salient content
> - Fine-Grain. = Fine-grained details
> - Rela. = Relational attributes
> - Peri. = Peripheral content
> - Faith. = Faithfulness
> - Know. = World knowledge
>
> This outcome strongly suggests that the model's perception capabilities are robust to the specific ordering of the CoD steps, as long as all descriptive components are present in the training data.
>
> We appreciate your valuable feedback and hope these responses address your concerns.

---

> > ### Author Response · Authors · 2025-11-27
> > **A Gentle Reminder**
> >
> > Dear Reviewer,
> >
> > Thank you once again for your valuable comments! As the discussion phase is approaching its end, we would like to kindly confirm whether we have sufficiently addressed all of your concerns (or at least some of them). Should there be any remaining questions requiring further clarification, please do not hesitate to let us know. We sincerely look forward to your feedback.

---

### Official Review · Reviewer_TmhP · 2025-11-01

**Soundness:** 3
**Presentation:** 3
**Contribution:** 3
**Rating:** 6
**Confidence:** 3

**Summary:**

This paper introduces SICOG (Self-Improving cognition), a self-learning framework designed to build next-generation foundation MLLMs by enhancing their cognitive abilities through pre-training with self-generated data. First, it proposes a self-improvement loop where a base model is fine-tuned with minimal supervision, then generates candidate captions and reasoning responses, which are filtered and used for its own large-scale pre-training. Second, it introduces "Chain-of-Description" (CoD), a novel structured mechanism that enables step-by-step visual understanding to improve perceptual quality , and integrates this with structured Chain-of-Thought (CoT) to enhance in-depth multimodal reasoning. Third, the framework uses a semantic-similarity-guided self-consistency mechanism to curate the high-quality, self-generated data, reducing reliance on external models. Finally, experiments demonstrate that this approach is effective, with SICOG achieving significant performance improvements on diverse multimodal benchmarks like MMStar and AI2D compared to previous pre-training methods.

**Strengths:**

- This paper is well written and easy to follow
- The proposed SICOG method is both simple and effective, enabling the model to bootstrap its own high-quality multimodal data without heavy external supervision.
- By injecting Chain-of-Description and structured Chain-of-Thought, the method produces rich, step-by-step image captions and reasoning chains, along with accurate VQA pairs.
- Comprehensive experiments on both low- and high-resolution backbones across 10+ standard benchmarks (MMStar, ScienceQA, AI2D, POPE, etc.) show consistent gains over leading pre-training approaches.

**Weaknesses:**

- A significant weakness of the SICOG paper is its naive curation filter, which risks systemic error propagation. The framework employs a "Self-Consistency-Guided Quality Evaluation" that selects the self-generated data candidate with the highest average semantic similarity to its peers, a method that fundamentally mistakes consensus for correctness. This creates a critical vulnerability: if the model has a consistent bias, such as a recurring hallucination , the filter will incorrectly reward this error as a "high-quality," high-consistency sample . This flawed data is then fed back into the pre-training loop (Stage 1.5) , reinforcing the very bias the framework is supposed to fix and leading to a potential "modal collapse" where the error becomes more entrenched over subsequent self-learning iterations.
- The framework's final training stage (Stage 1.5) uses a mix of self-generated perception data (CoD), reasoning data (CoT), and text-only data. The paper provides no analysis on how the ratio of these different data types impacts performance.

**Questions:**

See weaknesses.

---

> ### Author Response · Authors · 2025-11-17
> **Response to Reviewer TmhP**
>
> Thank you for your thoughtful and constructive feedback, which highlights the strengths of our work and provides valuable avenues for improvement. We address the specific points below.
>
> **Response to Weakness 1**:
>
> We appreciate your concern regarding error propagation from a consensus-based filter. To mitigate this, our framework incorporates an enhancement stage (Stage 1), where the base model is fine-tuned on a small, high-quality, human-annotated dataset before self-generation begins. This step improves the model’s perception and reasoning abilities, reducing the likelihood of systemic errors (e.g., recurring hallucinations) in self-generated outputs. By improving the model’s core competence upfront, the self-consistency filter operates on higher-quality, less biased candidates, reducing the risk of reinforcing errors. Appendix N demonstrates that this enhancement step leads to improved performance, highlighting its effectiveness. While we acknowledge that no filter is perfect, our approach minimizes bias risks and provides a strong foundation. We agree that exploring advanced filtering techniques is a valuable direction for future work.
>
> **Response to Weakness 2**:
>
> Thank you for highlighting this important point. Prevalent pre-training approaches primarily focus on image captioning datasets to acquire world knowledge. Our work argues that incorporating reasoning data (CoT) and text-only data during pre-training can significantly improve performance, particularly on multimodal reasoning tasks, as shown in Table 1. In Appendix G, we observe that scaling up self-generated captions without proportionally adjusting other data types leads to slight performance declines, highlighting the importance of maintaining a balanced data ratio rather than relying predominantly on image captioning data. While a detailed analysis of data ratios was beyond our computational resources, these findings establish a strong baseline. We acknowledge the value of further optimizing data ratios and plan to investigate this in future work.
>
> We appreciate your valuable feedback and hope these responses address your concerns.

---

> > ### Author Response · Authors · 2025-11-27
> > **A Gentle Reminder**
> >
> > Dear Reviewer,
> >
> > Thank you once again for your valuable comments! As the discussion phase is approaching its end, we would like to kindly confirm whether we have sufficiently addressed all of your concerns (or at least some of them). Should there be any remaining questions requiring further clarification, please do not hesitate to let us know. We sincerely look forward to your feedback.

---

### Author Response · Authors · 2025-11-28
**General Response**

We sincerely thank all reviewers for their thoughtful, detailed, and constructive feedback. We deeply appreciate the recognition of our contributions across multiple dimensions:

1. A novel self-improving multimodal learning paradigm that reduces dependence on external annotations. (TmhP, s2Hc, NPYP, WRjv)
2. Introduction of Chain-of-Description (CoD) for systematic visual perception and integration with structured CoT reasoning. (TmhP, s2Hc, NPYP, WRjv)
3. Comprehensive multimodal evaluation across 10+ benchmarks, demonstrating consistent gains with only 213K self-generated samples. (TmhP, s2Hc, NPYP, peVr)
4. Clear description of the full closed-loop framework and its practical advantages. (TmhP, s2Hc, NPYP, WRjv)

We also appreciate the reviewers’ efforts in highlighting important areas needing clarification or improvement. We have carefully addressed all concerns, including:

1. Error propagation risk in self-consistency filtering and model bias mitigation. (TmhP, s2Hc)
2. Role, necessity, and impact of Stage 1.5 pre-training. (NPYP, peVr)
3. Threshold selection and validation for semantic-similarity filtering. (NPYP)
4. Data contamination prevention and benchmark overlap checks. (NPYP)
5. Comparison with other data synthesis approaches and methodological distinction from prior self-evolution frameworks. (NPYP, peVr)
6. Sensitivity of CoD structure and ordering, including ablation studies. (s2Hc)
7. Computational efficiency and detailed GPU-hours analysis. (NPYP, peVr)
8. Generalization beyond standard benchmarks and avoidance of “feedback bubbles.” (WRjv)
9. Clarification on GRPO vs DPO and multi-turn conversational evaluation. (WRjv)
10. Organization improvements for appendix redundancy and theoretical exposition. (peVr)

All key clarifications, additional analyses, corrections (e.g., “35K”), and expanded results have been incorporated accordingly. All major modifications are highlighted in blue in the paper.

We sincerely thank all reviewers again for their time, insights, and constructive guidance.

---

### Note · Authors · 2025-12-27

I have read and agree with the venue's withdrawal policy on behalf of myself and my co-authors.